# Identification of an evolutionarily conserved domain in Neurod1 favouring enteroendocrine versus goblet cell fate

**Anne Sophie Reuter, David Stern**🔘**, Alice Bernard, Chiara Goossens**🔘**,
Arnaud Lavergne**🔘**, Lydie Flasse**¤**, Virginie Von Berg, Isabelle Manfroid**🔘**,
Bernard Peers**🔘**, Marianne L. Voz**🔘*

Laboratory of Zebrafish Development and Disease Models (ZDDM), GIGA, University of Liège, Liège,
Belgium

¤ Current address: Max Planck Institute of Molecular Cell Biology and Genetics (MPI-CBG), Dresden,
Germany
* mvoz@uliege.be

**Data Availability Statement:** All relevant data are
within the manuscript and its Supporting
Information files.

**Funding:** ASR, DS, AB and AL were supported by
FRIA (Fonds pour la Formation à la Recherche

## Abstract

ARP/ASCL transcription factors are key determinants of cell fate specification in a wide variety of tissues, coordinating the acquisition of generic cell fates and of specific subtype identities. How these factors, recognizing highly similar DNA motifs, display specific activities, is not yet fully understood. To address this issue, we overexpressed different ARP/ASCL factors in zebrafish *ascl1a-/-* mutant embryos to determine which ones are able to rescue the intestinal secretory lineage. We found that Ascl1a/b, Atoh1a/b and Neurod1 factors are all able to trigger the first step of the secretory regulatory cascade but distinct secretory cells are induced by these factors. Indeed, Neurod1 rescues the enteroendocrine lineage while Ascl1a/b and Atoh1a/b rescue the goblet cells. Gain-of-function experiments with Ascl1a/Neurod1 chimeric proteins revealed that the functional divergence is encoded by a 19-aa ultra-conserved element (UCE), present in all Neurod members but absent in the other ARP/ASCL proteins. Importantly, inserting the UCE into the Ascl1a protein reverses the rescuing capacity of this Ascl1a chimeric protein that cannot rescue the goblet cells anymore but can efficiently rescue the enteroendocrine cells. This novel domain acts indeed as a goblet cell fate repressor that inhibits *gfi1aa* expression, known to be important for goblet cell differentiation. Deleting the UCE domain of the endogenous Neurod1 protein leads to an increase in the number of goblet cells concomitant with a reduction of the enteroendocrine cells, phenotype also observed in the *neurod1* null mutant. This highlights the crucial function of the UCE domain for NeuroD1 activity in the intestine. As Gfi1 acts as a binary cell fate switch in several tissues where Neurod1 is also expressed, we can envision a similar role of the UCE in other tissues, allowing Neurod1 to repress Gfi1 to influence the balance between cell fates.

dans l'Industrie et dans l'Agriculture), CG is supported by the « Crédits Sectoriels de Recherche en Sciences de la Santé 2021/2022). LF was supported by the FNRS (https://www.frs-fnrs.be) and VVB by ARC-Uliege (Actions de Recherche Concertees) ((https://www.recherche.uliege.be). IM, BP, and MLV are associate researchers from FRS/FNRS (Fonds National pour la Recherche Scientifique). This work was funded by the "Credit de Recherche" (CDR) from the FNRS-FRS, the Léon Fredericq fund (https://www. fondationleonfredericq.be) and the Fonds Speciaux from the ULiège. The funders had no role in study design, data collection and analysis, decision to publish, or preparation of the manuscript.

**Competing interests:** The authors have declared that no competing interests exist.

## Author summary

It is not yet clear how highly related factors like the ARP/Ascl factors display specific activities even though they recognize the same consensus DNA motif. This specificity could be provided by their cellular environment or by intrinsic properties of the factors themselves. To distinguish between these two possibilities, we have expressed several ARP/Ascl factors in the *ascl1a-/-* mutant to determine which ones are able to rescue the intestinal secretory defects. We found that Ascl1a/b and Atoh1a/b are able to rescue the goblet cells while Neurod1 rescues the enteroendocrine lineage. Furthermore, we show that the specific Neurod1 activity is conferred by the presence of a 19-aa ultra-conserved element (UCE), present in all vertebrate Neurod members but absent in all the other ARP/ASCL proteins. This UCE domain, so far uncharacterized, acts as a goblet cell fate repressor and inhibits *gfi1aa* expression, known to be important for goblet cell differentiation. Inserting the UCE into Ascl1a protein reverses the rescuing capacity of this chimeric protein that cannot rescue the goblet cells anymore but can efficiently rescue the enteroendocrine cells. This study therefore highlights an unique intrinsic property of Neurod1 allowing it to repress Gfi1 to influence the balance between cell fates. As Gfi1 acts as a binary cell fate switch in several tissues where Neurod1 is also expressed, we can envision a similar role of the UCE in other tissues, allowing Neurod1 to repress Gfi1 to influence the balance between cell fates.

## Introduction

ARP/ASCL factors are key determinants of cell fate specification in a wide variety of tissues. Their role as cell fate determinants has been largely demonstrated in the developing nervous system where these factors are necessary and sufficient to confer a neural fate to progenitor cells [1]. These basic helix-loop-helix (bHLH) factors, named proneural factors, are members of the *Achaete scute-like* (ASCL) family or of the *atonal* related proteins (ARP) family, this latter family being further subdivided into Atonal (Atoh), Neurogenin (Neurog) and Neurod subfamilies. The proneural factors share functional properties as they coordinate the acquisition of generic neuronal fates [2]. However they also display specific functions; for example, ASCL1 is implicated in the generation and specification of GABAergic interneurons [3] while NEUROG2 is required for the generation of glutamatergic neurons in many regions of the central nervous system [2].

These proneural factors also act as cell fate determinants in the digestive system. For example, we and others have previously shown that Ascl1a is the cell fate determinant of the secretory lineage in zebrafish. Indeed, the *ascl1a^-/-^* mutant displays a complete loss of all secretory cells (i.e. the goblet and the enteroendocrine cells (EECs), the paneth cells being not found in zebrafish) [4,5]. Ascl1a is at the top of the secretory regulatory cascade as its knock-out leads to the loss of expression of all transcription factors known to be involved in secretory cell differentiation [5]. For example, this is the case for the transcription factor *sox4b*, detected in the gut nearly at the same time as *ascl1a* and co-expressed with *ascl1a* in all secretory progenitors. This is also true for *neurod1* which is expressed from 52 hpf onwards in all endocrine precursors. Intriguingly, and the role of cell fate determinant for the secretory lineage is played by another ARP/ASCL factors in the murine intestine, namely ATOH1 [6]. A similar evolutionary swap between ARP/ASCL factors is also observed in the pancreas where the cell fate determinant of the endocrine lineage is NEUROG3 in the murine pancreas [7] while this role is fulfilled by Ascl1b and Neurod1 in zebrafish [8]. Even within the same species, the identity of

the cell fate determinant can vary as for example, within the murine digestive system, ASCL1 is the determinant of the EEC lineage in the stomach [9] but is not expressed in the intestine [9,10] where this role is performed by NEUROG3 [11]. All these examples suggest that these ARP/ASCL factors share functional properties and are capable of fulfilling similar roles in different organs and species. These common properties could be explained by their binding to very similar E-box sequences via their highly related basic-helix-loop-helix DNA binding domain. Indeed comprehensive determination of protein-DNA specificities performed by Protein Binding Microarrays [12] revealed that *C.elegans* ARP/ASCL orthologues bind largely overlapping E-box motifs *in vitro* [13].

In this study, we addressed the question whether these ARP/ASCL factors share functional capacities and therefore would be interchangeable. We notably determined whether Atoh1 could replace Ascl1a as inducer of the secretory lineage in the zebrafish intestine. To that end, we induced Atoh1a or Atoh1b expression in the intestine of the *ascl1a-/-* embryos using HSP70 expression vectors and determined whether these factors could rescue the secretory defects observed in the mutant. We found that Atoh1a/b like Ascl1a/b are able to initiate the secretory cascade and that this property is also shared by Neurod1. However distinct secretory cells are induced by these factors as Ascl1a/b and Atoh1a/b rescue only the goblet cells while Neurod1 rescues exclusively the EEC lineage. By generating chimeric Ascl1a/Neurod1 proteins, we delineated the domain endowing Neurod1 with the ability to induce EECs. This domain, present in all vertebrate Neurod factors, is required for repressing the *gfi1aa* gene, thereby favouring EEC versus goblet cell fate. Deletion of this domain in the endogenous *neurod1* gene by CRISPR/Cas9 editing underlined its importance in the secretory cell fate specification.

## Results

### Atoh1 and Neurod1, like Ascl1, are able to initiate the intestinal secretory lineage in the *ascl1a-/-* larvae

As Ascl1a is the cell fate determinant of the secretory cells in zebrafish and Atoh1 in mouse [5,6], we determined whether these two factors have the same biological activity and can be swapped. We therefore tested whether the zebrafish orthologues Atoh1a and Atoh1b could rescue the secretory defects of the *ascl1a-/-* mutant by performing gain-of-function (GOF) experiments. As a prerequisite, we verified whether inducing Ascl1a itself in *ascl1a-/-* larvae could rescue the expression of *sox4b*, known to be directly downstream of Ascl1a in the secretory regulatory cascade [5]. To that end, we generated a transgene *Tg(hsp70l:eGFP-2A-ascl1a)*, where the heat-shock inducible promoter *hsp70l* controls the expression of a bicistronic transcript, eGFP-2A-Ascl1a. To mimic endogenous *ascl1a* gene expression that appears around 36 hpf in the gut [5], we performed two successive heat shocks at 36 and 46 hpf that should maintain transgene expression for at least 20 hours. This leads to the full rescue of *sox4b* expression at 55 hpf in all *ascl1a-/-* mutant larvae (Fig 1C, n = 7). The same results were obtained when inducing the paralog *ascl1b* (S1B Fig). Similarly, full rescue of *sox4*b expression was also observed with Atoh1b or Atoh1a using *Tg(hsp70l:atoh1b-Myc)* [14] and *Tg(hsp70l:atoh1a)* [15] (Fig 1D, n = 15 and S1C Fig), indicating that Atoh1a and Atoh1b are both able to initiate the secretory transcriptional cascade like Ascl1a. To determine whether this property could be extended to the third group of the ARP/ASCL family, we constructed the transgene *Tg(hsp70l: eGFP-2A-neurod1)* and found that Neurod1 is also able to fully rescue *sox4*b expression in all *ascl1a-/-* larvae (Fig 1E, n = 16). In conclusion, these data indicate that Atoh1a/b, Ascl1b and Neurod1 share with Ascl1a the capacity to initiate the intestinal secretory cascade, highlighting common biological activities between these factors.

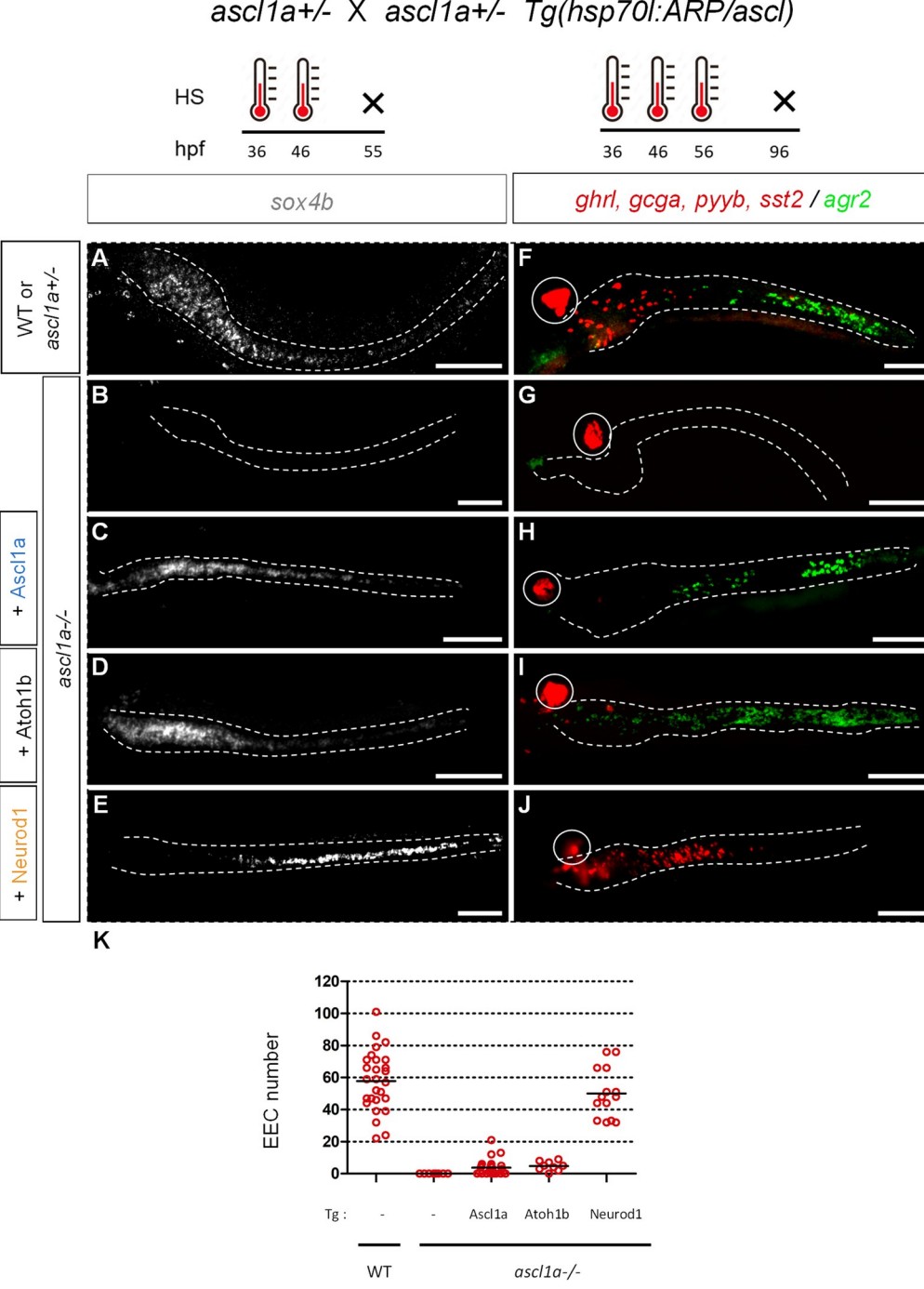

**Fig 1. Rescue of *ascl1a-/-* by ARP/Ascl factors.** Upper part: scheme of the crossings and timing of the heat shocks (HS). Lower part: **A-E**: FISH performed at 55 hpf with a *sox4b* probe on *ascl1a^-/-^* or control sibling embryos heat-shocked at 36 and 46 hpf **F-J**: FISH performed at 96 hpf with the *agr2* probe revealed in green and a mix of hormone probes (*ghrelin (ghrl)*, *peptide YYb (pyyb)*, *glucagon-a (gcga)* and *somatostatin-2 (sst2))* revealed in red on *ascl1a^-/-^* or control sibling embryos heat-shocked at 36, 46 and 56 hpf. The transgenic line used is indicated on the left part of the figure as well as the genotype of the larvae; the ascl1a^-/-^ larvae were identified by the loss of the pituitary *prl* expression (not shown). The pancreas is encircled while the location of gut, visualised with a DAPI staining (not shown), is delimited by dashed lines. **K**: Quantification of the number of *ghrl+/pyyb+/gcga+/sst2+* EEC detected in conditions F to J counted under a fluorescent microscope. All views are ventral with the anterior part to the left and represent confocal projection images. Scale bar: 100 μm.

## Neurod1 is able to rescue the EEC lineage of *ascl1a-/-* larvae while Atoh1b and Ascl1a rescue the goblet cell fate

The next step was to determine whether the rescued *sox4b+* secretory progenitor cells could pursue their differentiation process and give rise to all types of secretory cells (i.e. goblet and EECs). For that purpose, we performed 3 successive heat shocks at 36, 46 and 56 hpf as we determined that it was the optimal rescue conditions. We checked the expression at 96 hpf of the goblet marker *agr2* and of several enteroendocrine hormones. These latter were monitored by FISH using a mix of hormone probes (*ghrelin (ghrl)*, *glucagon-a (gcga)*, *peptide YY (pyyb)*, and *somatostin-2 (sst2)*). Surprisingly, Ascl1a and Atoh1b rescued the formation of numerous goblet cells and of only a very few EECs (Fig 1H–1I; quantification for EE on 1K) and the same results were obtained with their paralogs Ascl1b and Atoh1a (S1D–S1I Fig). In contrast, Neurod1 rescued exclusively the EEC lineage (Fig 1J) at a level similar to wild-type larvae (Fig 1F and 1K). By testing the hormone probes separately, we observed that Neurod1 is able to rescue all tested enteroendocrine hormones, i.e. *adenylate cyclase-activating polypeptide 1* (*adcyap1a*), *gcga*, *insulin-like peptide 5a* (*insl5a*) and *pyyb* (S2 Fig). To determine why Ascl1a and Atoh1b fail to restore EECs in *ascl1a-/-* mutants, we assessed the expression of *neurod1*, one of the earliest transcription factors expressed in all endocrine progenitors. We found that heat-shock induced expression of *ascl1a* and *atoh1b* could not rescue the expression of *neurod1* (S3C and S3D Fig). In contrast, heat-shock induced expression of *neurod1* triggered the expression of the endogenous *neurod1* gene (S3E Fig). This was done using a probe localised in the 3'UTR of *neurod1* that does not recognize the *Tg(hsp70l:eGFP-2A-neurod1)* transgene. This suggests that Neurod1 is essential for EEC generation and that the incapacity of Ascl1a or Atoh1b to induce neurod1 prevent them to rescue the EEC lineage.

The fact that Ascl1a is not able to rescue the EECs in *ascl1a-/-* mutants is at first sight surprising. This is most likely due to the HSP70 system which leads to an ubiquitous expression of the *ascl1a* transgene while the endogenous *ascl1a* gene is expressed in scattered cells within the intestinal epithelium. This probably disturbs signalling pathways, such as the Notch signalling pathway, important for an optimal differentiation of these cells (see discussion). Nevertheless, inducing these three ARP/Ascl factors in the same context highlights different intrinsic properties of Neurod1 compared to Ascl1a and Atoh1: Neurod1 GOF rescue the EEC lineage in *ascl1a-/-* mutants while Ascl1a/b and Atoh1a/b GOF rescue the goblet cells.

## Neurod1 contains an evolutionary conserved domain that represses goblet cell fate

To identify the domain(s) of Neurod1 conferring its ability to efficiently rescue the EECs, we swapped different functional domains between Neurod1 and Ascl1a and tested the ability of the chimeric proteins to rescue secretory lineages in the *ascl1a-/-* mutant. Fig 2 displays the structure of the different chimeric proteins (Tg1 to Tg8) with a summary of their biological activities as shown in detail below. Replacing the N-terminal domain containing the basic domain (b) or the bHLH of Neurod1 by the corresponding region of Ascl1a (constructs Tg3: b$^A$-HLH-Cterm$^{ND}$ and Tg4: bHLH$^A$-Cterm$^{ND}$) results in proteins with rescuing capacities equivalent to those of the wild-type Neurod1 protein i.e. able to rescue the EECs in *ascl1a-/-* larvae but not the *agr2+* goblet cells (Fig 3D and 3E, quantification on 3H). In contrast, the Tg5 (b$^{ND}$-HLH-Cterm$^A$) construct, which contains the basic domain of Neurod1 and the HLH and the C-terminal domain of Ascl1a, displays similar properties as wild-type Ascl1a i.e. the capacity to rescue goblet cells but not EECs (compare Fig 3F with 3B). All these data, summarized in Fig 2 (column 2 and 3, lanes Tg1 to Tg5), indicate that the elements crucial for rescuing EECs are present in the C-terminal region of Neurod1. Protein sequence alignment of

| | | | In *ascl1a-/-* , rescue of | | | In WT |
| --- | --- | --- | --- | --- | --- | --- |
| | | | 1) *sox4b+* cells | 2) EE cells | 3) Goblet cells | 4) Early ectopic induction of *pax6b+* cells |
| Tg1 | *ascl1a* | | ✓ | 8 ± 0,9 | 86% | 1% |
| Tg2 | *neurod1* | | ✓ | 100 ± 6,7 | 0% | 100% |
| Tg3 | $b^A$ *HLH-C-term*$^{ND}$ | | ✓ | 56 ± 4,9 | 0% | 59% |
| Tg4 | *bHLH*$^A$ *C-term*$^{ND}$ | | ✓ | 86 ± 5,8 | 0% | N.D. |
| Tg5 | $b^{ND}$ *HLH-C-term*$^A$ | | ✓ | 18 ± 1,2 | 75% | 2% |
| Tg6 | *ascl1a insECD*$^{ND}$ | | ✓ | 32 ± 6,7 | 0% | 30% |
| Tg7 | *neurod1ΔUCE* | | ✓ | 70 ± 7,7 | 100% | 35% |
| Tg8 | *neurod1ΔECD* | | ✓ | 33 ± 5,2 | 100% | N.D. |

**Fig 2. Summary of the rescuing and inducing capacities of Ascl1a, Neurod1 and the chimeric Ascl1a/Neurod1 proteins. Left part**: Schematic representation of the different transgenes used in this study. **Column 1**: As indicated by a √ sign, Tg1 to Tg8 are all able to rescue *sox4b* at 55 hpf in the *ascl1a-/-* larvae, heat-shocked at 36 and 46 hpf (data from Fig 1). **Column 2**: Comparison of EEC rescuing capacities of Tg1 to Tg8, compared to Tg2 (Neurod1), arbitrarily set to 100%. *ascl1a-/-* larvae were heat-shocked at 36, 46 and 56 hpf and the number of EECs determined at 96 hpf by FISH. Data from Figs 3 and 5. **Column 3**: Percentage of *ascl1a-/-* larvae showing a rescue of goblet cells at 96 hpf upon the expression of Tg1 to Tg8, induced by 3 heat-shocks at 36, 46 and 56 hpf. Data from Figs 3 and 5 **Column 4**: Comparison of capacities of Tg1 to Tg8 to induce *pax6b+* cells in wt larvae compared to Tg2 (Neurod1), arbitrarily set to 100%. Larvae were heat-shocked at 48 and 58 hpf and the number of *pax6b+* cells determined at 56 hpf by FISH. Data from S4 Fig. N.D.: not done.

this region from different vertebrate species reveals the presence of an evolutionary conserved domain of 41 amino-acids (aa) which is present in all vertebrate Neurod family members (Fig 4) but is not found in the other ARP/Ascl factors. This domain, whose function is so far unknown, was named ECD for Evolutionarily Conserved Domain. The ECD domain, directly flanked by the bHLH on its N-terminal side and by the transactivation domains on its C-terminal side (Fig 4), contains in its centre a 19-aa ultra-conserved element (UCE). Interestingly, we found that the insertion of this ECD domain in Ascl1a (Tg6) completely abolishes its capacity to rescue the goblet cells and, inversely, allows the rescue of EECs (Fig 5B–5B', quantification on 5E–5F). These data demonstrate that the ECD is a crucial region for the specific biological activity of Neurod1 that represses goblet cell fate. This was further confirmed by deleting the ECD or the smaller UCE domain in Neurod1 (Neurod1$^{\Delta ECD}$ (Tg8) or Neurod1$^{\Delta UCE}$ (Tg7)) that lead to a full rescue of the goblet cells (Fig 5C', 5D' and 5F) while these two transgenes were still able to trigger the formation of EECs (Fig 5C, 5D and 5E).

All these data, summarized in Fig 2 (columns 2 and 3), indicate that the UCE of Neurod1 functions primarily as a repressor of goblet cell fate in the context of intestinal cell differentiation.

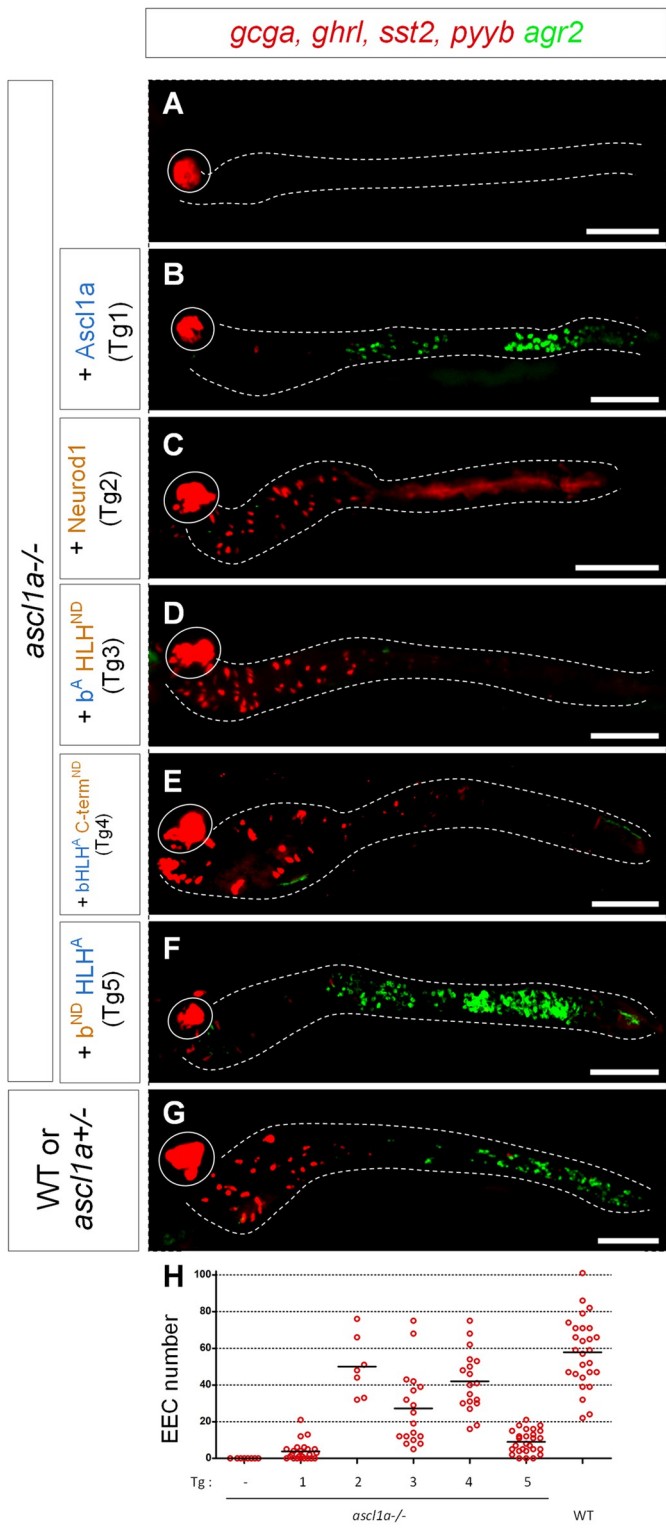

**Fig 3. Analysis of rescuing capacities of chimeric Neurod1/ Ascl1a proteins in *ascl1a*-/- larvae. A-G:** FISH performed at 96 hpf with the *agr2* probe revealed in green and a mix of hormones probes (*ghrl*, *pyyb*, *gcga and sst2*) revealed in red on *ascl1a* -/- larvae without (A) or with Tg1 to Tg5 transgenes (B to F) and on sibling control embryos (G). All the larvae have been heat-shocked at 36, 46 and 56 hpf. The transgenic line used is indicated on the left part of the figure as well as the genotype of the larvae; the *ascl1a*$^{-/-}$ larvae were identified by the loss of the pituitary *prl*

expression (not shown). All views are ventral with the anterior part to the left and represent confocal projection images. The pancreas is encircled while the location of the gut, visualised with a DAPI staining (not shown), is delimited by dashed lines. Scale bar: 100µm. **H:** Quantification of the number of *ghrl+/pyyb+/gcga+/sst2+* EEC detected in conditions A to G counted under a fluorescent microscope.

### RNAseq transcriptome profiling of the genes regulated by Neurod1, Neurod1$^{\Delta ECD}$, Neurod1$^{\Delta UCE}$ and Ascl1a

To further decipher Neurod1 mechanisms of action and the role of its ECD domain, we next determined by RNA-seq the genes regulated by Neurod1 and its deleted forms Neurod1$^{\Delta ECD}$ and Neurod1$^{\Delta UCE}$ compared to Ascl1a. Ideally, such experiments should be performed in the *ascl1a-/-* larvae as described below; however, these experiments are hard to implement as *ascl1a-/-* larvae cannot be morphologically distinguished from their sibling larvae (*ascl1a-/+* and +/+) and as homozygous mutants die before reaching adulthood. To circumvent this problem, we took advantage of the fact that the overexpression of Neurod1 in wild-type embryos through heat shocks at 38 and 48 hpf induces the pan-endocrine marker *pax6b* in the gut at 56 hpf (S4C Fig). At that stage pax6b is not yet detected in the gut of non-transgenic embryos as *pax6b* expression onset in the intestine is around 60 hpf [16] (S4A Fig). Furthermore, we observed that constructs Tg3, Tg6 and Tg7 were also able to induce *pax6b* expression earlier albeit at lower levels compared to Neurod1 (S4D, S4F and S4G Fig, quantification on

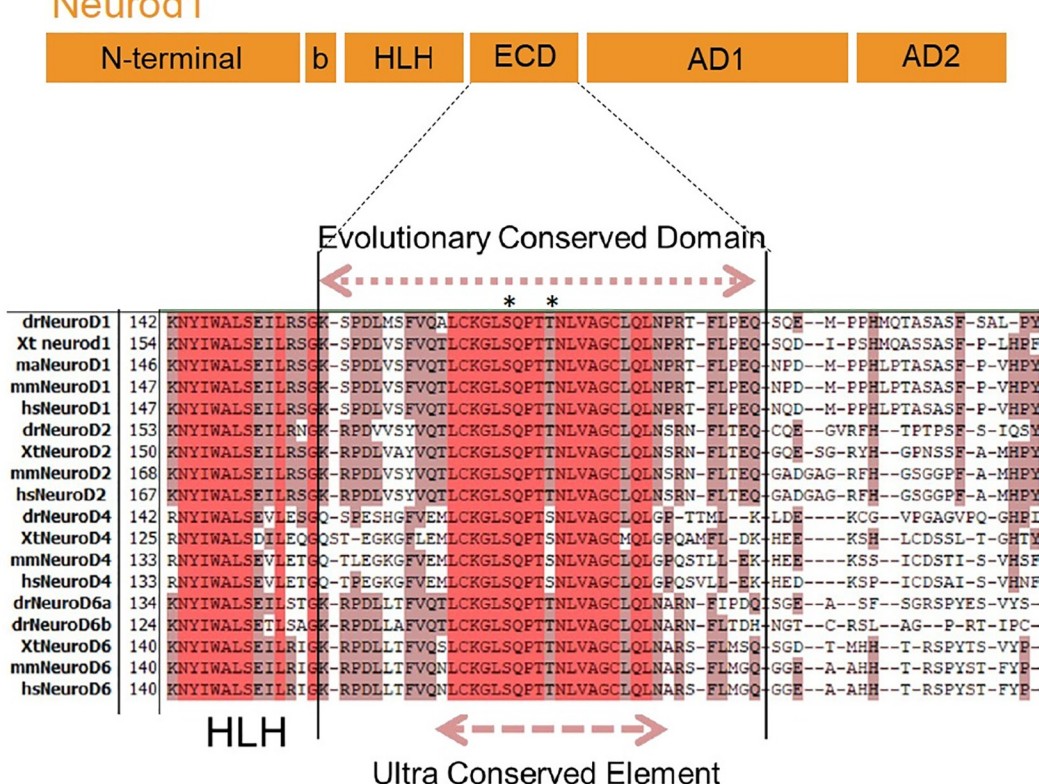

**Fig 4. Neurod1 contains a domain evolutionarily conserved in all vertebrate Neurod family members. Upper part:** Schematic representation of dr-Neurod1 showing the 12aa basic domain (b), the 45aa helix-loop-helix (HLH) domain, the 41aa evolutionary conserved domain (ECD) and the two transactivation domains (AD1 and AD2). **Lower part**: Alignment of vertebrate Neurod proteins highlighting the ECD and the 19aa ultra conserved elements (UCE). The potential phosphorylation site for the Serine/threonine protein kinases ATM/ATR/DNA-PK (S172 in dr-Neurod1) is indicated by an asterisk. The presence of a conserved S or T at aa176 in dr-Neurod1, highly suggestive of a site of phosphorylation for an unidentified serine/threonine protein kinase, is also indicated by an asterisk.

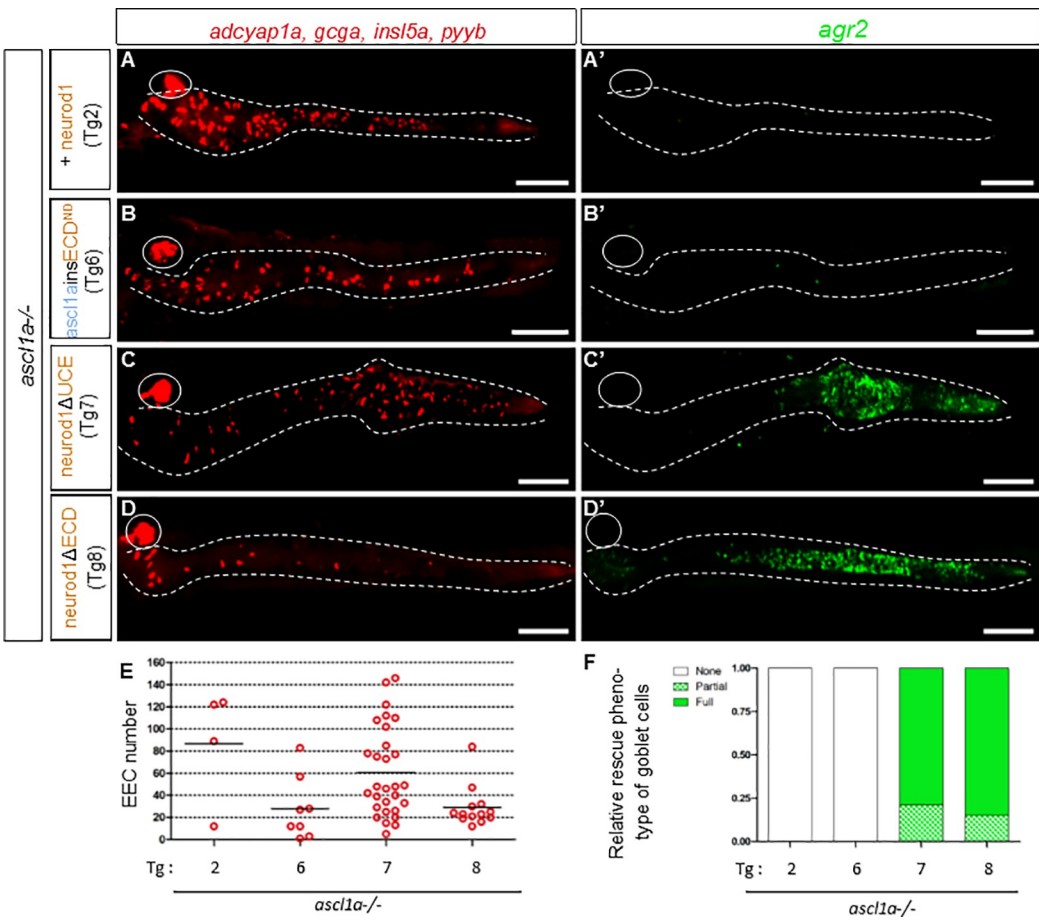

**Fig 5. The ECD domain of Neurod1 represses goblet cell fate. A-D**: FISH performed at 96hpf with a mix of hormones probes (*adcyap1a*, *gcga*, *insulin-like5a (insl5a) and pyyb*) revealed in red and with the *agr2* probe revealed in green (**A'-D'**) on *ascl1a-/-* transgenic embryos heat-shocked at 36, 46 and 56hpf. The transgenic line used is indicated on the left part of the figure. All views are ventral with the anterior part to the left and represent confocal projection images. The pancreas is encircled while the location of the gut, visualised with a DAPI staining (not shown), is delimited by dashed lines Scale bar: 100 μm. **E:** Number of EEC rescued cells in the gut when inducing the different transgenes (Tg2, Tg6 to Tg8) in *ascl1a-/-* larvae **F:** Percentage of larvae showing no, partial or full rescue of goblet cells when inducing the different transgenes (Tg2, Tg6 to Tg8) in *ascl1a-/-* larvae. The larvae expressing *agr2* at a level similar to wt or higher were classified as "Full rescue" while larvae with lower expression of *agr2* compared to wt were classified as "partial rescue". Larvae without *agr2* expression were classified as "no rescue".

S4H). This is concordant with the reduced capacity of these chimeric constructs to restore EECs in *ascl1a-/-* mutant embryos (Fig 2, compare columns 2 and 4). In contrast, the Tg1 (Ascl1a) or Tg5 transgenes, not able to rescue EEC in *ascl1a-/-*, are not able either to induce *pax6b+* cells (S4B and S4E Fig). All these data, summarized on Fig 2 (column 4), indicate that RNA-seq profiling on the gut of wild-type transgenic embryos should allow the identification of genes specifically regulated by Neurod1 and its deleted forms. As the induction of *pax6b* expression was already observed at 52 hpf in about 75% of the embryos, we performed the RNA-seq at this stage in order to minimize the number of indirect targets of Neurod1.

Practically, the endodermal reporter line Tg(*sox17*:*dsred*) was crossed with the transgenic lines for Ascl1a (Tg1), Neurod1 (Tg2), Neurod1$^{\Delta UCE}$ (Tg7) and Neurod1$^{\Delta ECD}$(Tg8). The double transgenic embryos were heat-shocked at 38 and 48 hpf as well as the simple transgenic Tg (*sox17*:*dsred*) embryos used as control. At 52 hpf, the trunk of about 150 embryos was dissected, the cells dissociated and the *dsred+* endodermal cells isolated by Fluorescence-Activated

Cells Sorting (FACS). The RNA-seq were performed at least in triplicates for each construct as previously described [16].

The comparison of the RNA-seq data revealed 104 differentially expressed (DE) genes upon overexpression of Neurod1 compared to control embryos, 79 induced and 25 repressed genes) and 92 Ascl1a-regulated genes (21 induced and 71 repressed genes, False discovery rate (FDR) ≤ 0,1) (S1A and S1B Table). The direct comparison of Neurod1- versus Ascl1a-induced samples identifies 98 additional DE genes (Fig 6A). These 279 DE genes, listed in S1C Table, were

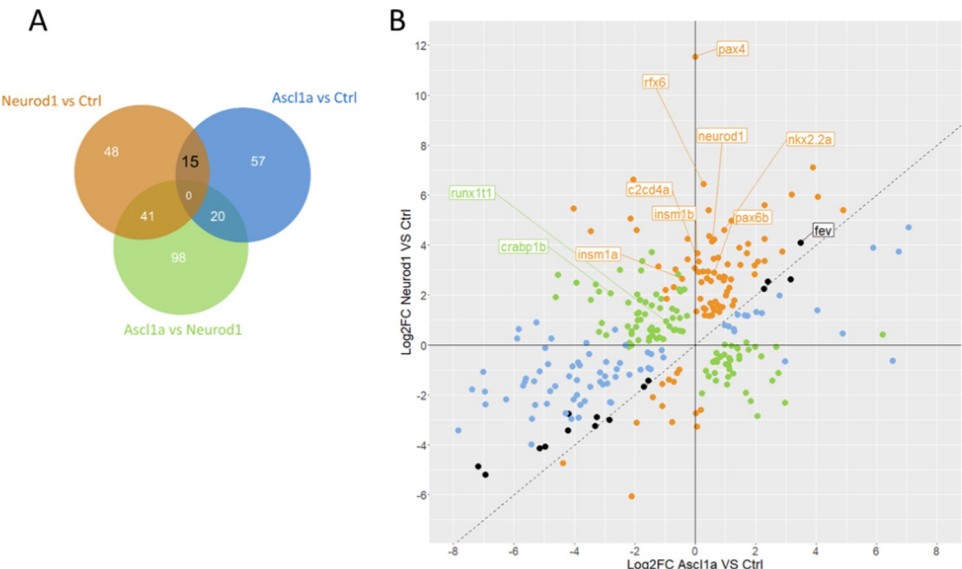

| | Gene name | Mean Ctrl | Mean tg1 (ascl1a) | fold change (tg1 vs Ctrl) | Mean tg2 (neurod1) | fold change (tg2 vs Ctrl) | Mean tg7 (ΔUCE) | fold change tg7 vs Ctrl | Mean tg8 (ΔECD) | fold change (tg8 vs Ctrl) |
|---|---|---|---|---|---|---|---|---|---|---|
| 1 | pax4 | 0.0 | 0.0 | 1.0 | 46.6 | 47.6 | 17.0 | 18.0 | 17.5 | 18.5 |
| | rfx6 | 0.4 | 0.4 | 1.1 | 32.0 | 24.5 | 10.4 | 8.4 | 4.0 | 3.7 |
| | neurod1 (3' UTR)* | 2.7 | 3.6 | 1.4 | 45.7 | 17.5 | 36.9 | 13.5 | 6.1 | 4.2 |
| | nkx2.2a | 1.6 | 3.0 | 1.5 | 41.2 | 16.1 | 9.7 | 4.1 | 10.1 | 4.2 |
| | insm1b | 20.0 | 19.6 | 1.0 | 256.3 | 12.3 | 115.4 | 5.5 | 105.1 | 5.1 |
| | insm1a | 20.1 | 13.9 | 0.7 | 126.7 | 6.0 | 52.5 | 2.5 | 35.2 | 1.7 |
| | c2cd4a | 1.0 | 1.1 | 1.0 | 10.6 | 5.7 | 6.3 | 3.6 | 3.1 | 2.0 |
| | pax6b | 2.9 | 4.2 | 1.4 | 21.0 | 5.7 | 9.0 | 2.6 | 7.0 | 2.1 |
| | sox4b | 39.4 | 73.6 | 1.8 | 174.1 | 4.3 | 158.3 | 3.9 | 102.0 | 2.5 |
| | tm4sf4 | 154.4 | 289.5 | 1.9 | 608.2 | 3.9 | 794.1 | 5.1 | 548.9 | 3.5 |
| | | | | 1.3 | | 14.4 | | 6.7 | | 4.8 |
| 2 | crabp1b | 35.0 | 4.2 | 0.1 | 138.9 | 4.2 | 100.2 | 3.5 | 96.7 | 3.2 |
| | greb1l | 90.0 | 55.4 | 0.7 | 164.4 | 1.8 | 135.9 | 1.5 | 88.5 | 1.1 |
| | runx1t1 | 11.0 | 5.3 | 0.5 | 21.9 | 1.9 | 27.4 | 2.5 | 15.6 | 1.5 |
| | dlb | 26.6 | 5.2 | 0.2 | 87.5 | 3.3 | 39.6 | 1.5 | 50.0 | 1.9 |
| | dlc | 38.2 | 50.7 | 1.3 | 85.9 | 2.2 | 61.9 | 1.6 | 69.5 | 1.8 |
| | dll4 | 5.1 | 2.0 | 0.4 | 46.6 | 9.1 | 13.3 | 2.6 | 22.4 | 4.4 |
| 3 | gfi1aa | 1.0 | 2.5 | 2.5 | 5.2 | 5.3 | 59.7 | 60.2 | 37.3 | 37.6 |
| | plcxd3 | 0.3 | 0.0 | 0.0 | 13.6 | 47.0 | 0.2 | 0.6 | 0.4 | 1.5 |
| | cacna1ba | 16.6 | 9.3 | 0.6 | 1.9 | 0.1 | 29.4 | 1.8 | 17.9 | 1.1 |
| | mfsd4aa | 3.5 | 4.1 | 1.2 | 6.1 | 1.7 | 36.0 | 10.2 | 25.4 | 7.2 |

**Fig 6. Comparison of the differentially expressed genes. A:** Venn diagram showing the differentially expressed genes for Neurod1 (Tg2) and for ascl1a (Tg1) compared to Ctrl and for Ascl1a vs Neurod1 (FDR ≤ 0,1). **B**: Plot showing the log2 Fold Change after overexpression of Ascl1a (Tg1) (x-axis) and Neurod1 (Tg2) (y-axis) compared to Ctrl embryos for the 209 DE genes. Genes in blue represent significant Ascl1a DE, in orange, Neurod1 DE and genes in black correspond to genes significantly DE in both conditions. Genes in green represent the 98 additional DE genes identified through the direct comparison of Neurod1- versus Ascl1a-induced samples. **C:** Expression level of selected DE genes. The expression level (given in normalized CPM) was obtained from the RNA-seq data from Ctrl or hsp70l transgenic lines for Ascl1a (Tg1), Neurod1 (Tg2), Neurod1$^{\Delta UCE}$ (Tg7) and Neurod1$^{\Delta ECD}$ (Tg8) embryos, all heat-shocked at 38 and 48 hpf and harvested at 52 hpf. The values are the expression mean of at least triplicate samples. **Panel 1:** Expression level for the 10 "endocrine pancreas development" genes. **Panel 2**: Expression level of selected genes differentially regulated by Neurod1 compared to Ascl1a. **Panel 3**. Expression level of selected genes differentially regulated by Neurod1$^{\Delta UCE}$ and Neurod1$^{\Delta ECD}$ compared to Neurod1.

plotted according to the level of regulation by Neurod1 or Ascl1a (Fig 6B). This figure highlights that most of the DE genes regulated by Neurod1 (in orange) are induced while Ascl1a-DE genes (in blue) are mainly repressed by Ascl1a. All these comparisons highlighted several genes regulated only by NeuroD1, such as *pax4*, *rfx6*, *insm1a/b*, genes reported to be direct targets of Neurod1 in human insulinoma cell line [17], therefore validating our experiments (Fig 6B). GO analyses of Neurod1-regulated genes compared to wild-type reveal that the highest enrichment is seen for "endocrine pancreas development" with a list of 10 genes (FDR = 1.1 E-11). This endocrine gene set includes in addition to *pax4*, *rfx6* and *insm1a/b*, *pax6b*, *nkx2.2a*, *c2cd4a* and *tmsf4a* (Fig 6C, panel 1). Identifying pancreatic genes expressed in the intestine is not surprising as pancreatic and intestinal endocrine cells share regulatory programs [16]. Neurod1 overexpression also significantly induces the expression of the endogenous *neurod1* gene as visualized by the drastic increase (17.5-fold) of the reads spanning the 3'UTR of *neurod1* (not present in the Tg2 construct). As shown in Fig 6C, panel 1, these 10 genes are not induced by Ascl1a, confirming the unique capacity of Neurod1 to efficiently induce the EE program. The comparison of Ascl1a and Neurod1 regulated genes versus Ctrl reveals a small overlap of 15 genes (Fig 6A), all being regulated in the same way and with a similar amplitude by Neurod1 and Ascl1a (Fig 6B, genes shown as black dots). This common gene set includes *fev*, a transcription factor highly induced by both factors (>10 fold). Fev has been reported to be highly expressed at early stage in the EEC lineage that will give rise to the Tac1+ cells [18]. The direct comparison of Ascl1a and Neurod1 RNA-seq data identifies 159 DE genes that includes several DE identified above (61 genes) but also highlights additional genes which are mostly induced by Neurod1 and repressed by Ascl1a (Fig 6B, green dots). This is the case notably for *crabp1* and *greb1l*, two important actors for retinoic acid signalling and for *runx1t1*, involved in the EEC differentiation of K, I and D lineage[18] (Fig 6C, panel 2). This list includes also several Notch ligands as Neurod1 activate *deltaB (dlb)*, *deltaC (dlc)* and *delta-like 4 (dll4)* while Ascl1a represses the expression of *dlb* and *dll4* suggesting an inverse regulation of Delta/Notch signalling by Ascl1a and Neurod1.

Regarding the deleted versions Neurod1$^{\Delta UCE}$ (Tg7) and Neurod1$^{\Delta ECD}$ (Tg8), both are still able to induce the expression of the 10 genes associated with the "endocrine pancreas development" signature, though less efficiently than the full length Neurod1 (Tg2) (Fig 6C, panel 1). Indeed, the fold induction of each gene is lower for the deleted versions than for Tg2, the mean induction for these 10 genes being 14.4-fold for Neurod1, 6.7-fold for the deleted construct Neurod1$^{\Delta UCE}$ and 4.8-fold for the larger deletion Neurod1$^{\Delta ECD}$ (by comparison, this fold change is 1.3 fold for Ascl1a). The comparison of Neurod1$^{\Delta UCE}$ with Neurod1$^{\Delta ECD}$ RNA-seq data identifies only one DE gene between these 2 conditions (the unknown BX548062.1 gene). This strongly suggests that these deleted mutant proteins have similar transcriptional properties. Therefore, we next compared the combined RNA-seq data for Neurod1$^{\Delta UCE}$ and Neurod1$^{\Delta ECD}$ with Neurod1 and identified 28 DE genes (S1E Table). Among them, 21 genes are induced by Neurod1$^{\Delta UCE}$ and Neurod1$^{\Delta ECD}$ but not by Neurod1. Gfi1aa (Growth factor independence 1aa) is the most significant one with an upregulation of 60- and 38-fold in Neurod1$^{\Delta UCE}$ and Neurod1$^{\Delta ECD}$, respectively (Fig 6C, panel 3). In mice, G*fi1* is expressed in goblet cells and is essential for their generation [19]. In zebrafish, *gfi1aa:GFP* drives GFP expression in the goblet cells [20]. To validate these RNA-seq data, we next performed *in situ* hybridization. We determined whether *gfi1aa* expression was induced in the gut by Neurod1$^{\Delta UCE}$ or Neurod1$^{\Delta ECD}$ GOF. We found that inducing Neurod1$^{\Delta UCE}$ in wild-type embryos by one heat shock at 48 hpf triggered high *gfi1aa* expression in the gut of most embryos (18 out 20) (Fig 7A) while its expression was not detected in control embryos at this stage (n = 4) (Fig 7B). This is also the case for all Neurod1$^{\Delta ECD}$ GOF embryos (n = 5) albeit at a lesser extent (Fig 7C),

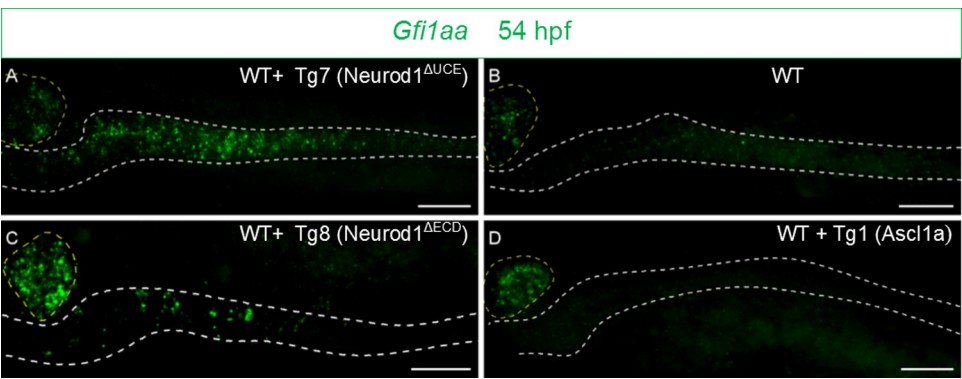

**Fig 7. Intestinal *gfi1aa* expression is highly induced by Neurod1$^{\Delta UCE}$ and Neurod1$^{\Delta ECD}$.** FISH performed with the *gfi1aa* probe on wild-type **(B)** embryos or carrying the Tg7 (Neurod1$^{\Delta UCE}$) **(A)**, Tg8 (Neurod1$^{\Delta ECD}$)**(C)** or Tg1 (Ascl1a) **(D)** transgenes. All the embryos have been heat-shocked at 48 hpf and analysed at 54 hpf. All views are lateral with the anterior part to the left and represent confocal projection images. The pancreas is encircled with yellow dashed lines while the location of the gut, visualised with a DAPI staining (not shown), is delimited by white dashed lines. Scale Bars: 50 μm.

while *gfi1aa* was not induced upon Ascl1a GOF (n = 3) (Fig 7D), confirming therefore the RNAseq data.

In conclusion, all these RNA-seq data show that Neurod1 and Ascl1a regulate mostly different sets of genes and that the UCE domain is required for the optimal induction of the genes specifically induced by Neurod1. The UCE domain is also required for the repression of the *gfi1aa* gene whose murine orthologue is known to be involved in goblet cell differentiation.

## Neurod1$^{\Delta UCE}$ displays an increase of goblet cells and a concomitant reduction of EECs

To validate our GOF experiments indicating an important role of the Neurod1 UCE domain in the secretory cell fate choice, we deleted the UCE in the endogenous *neurod1* gene by crispr/cas9 mutagenesis. We generated the mutant line *neurod1$^{ulg052}$*, called here *neurod1$^{\Delta UCE}$*, harboring a deletion of 81 bp that includes the 19aa of the UCE flanked by 3 and 5 aa of the ECD. The number of goblet cells was determined in *neurod1$^{\Delta UCE}$* by labelling with rhodamine dextran-conjugated wheat germ agglutinin (WGA) that interacts with the N-acetylglucosamine of the mucus [21]. A significant increase in the number of goblet cells was detected in the intestine of *neurod1$^{\Delta UCE}$* compared to wild-type embryos at 5 dpf (Fig 8A–8C), confirming that the UCE domain of Neurod1 is a repressor of the goblet lineage. This increase of goblet cells was also visualised by an alcian blue staining of microtome serial sections (Fig 8L–8M). Based on our GOF experiments, the deletion of the UCE domain should also reduce the capacity of Neurod1 to induce the endocrine program. To visualise more easily the EECs, the *neurod1$^{\Delta UCE}$* mutants were crossed with the transgenic *Tg(pax6b:GFP)$^{ulg515Tg}$* line [22] where all EECs are fluorescently labelled. A two-fold reduction of *pax6b:GFP* expression was observed in the *neurod1$^{\Delta UCE}$* homozygous mutants at 4 and 5 dpf (Fig 8D–8I) confirming the importance of the UCE for inducing the EE program. We analysed further the *neurod1$^{\Delta UCE}$* phenotype by determining the EEC transcriptome at 4 dpf of *Tg(pax6b:GFP)/neurod1$^{\Delta UCE}$* larvae. *GFP*+ EECs were obtained by microdissection of the gut followed by cell dissociation and selection of *GFP*+ by FACS. RNA-seq was performed on 4 independent homozygous *neurod1$^{\Delta UCE}$* and 6 wild-type EEC preparations. The analyses of the RNA-seq data revealed 680 DE genes between *neurod1$^{\Delta UCE}$* homozygous mutants and ctrl larvae (FDR ≤ 0,1, average

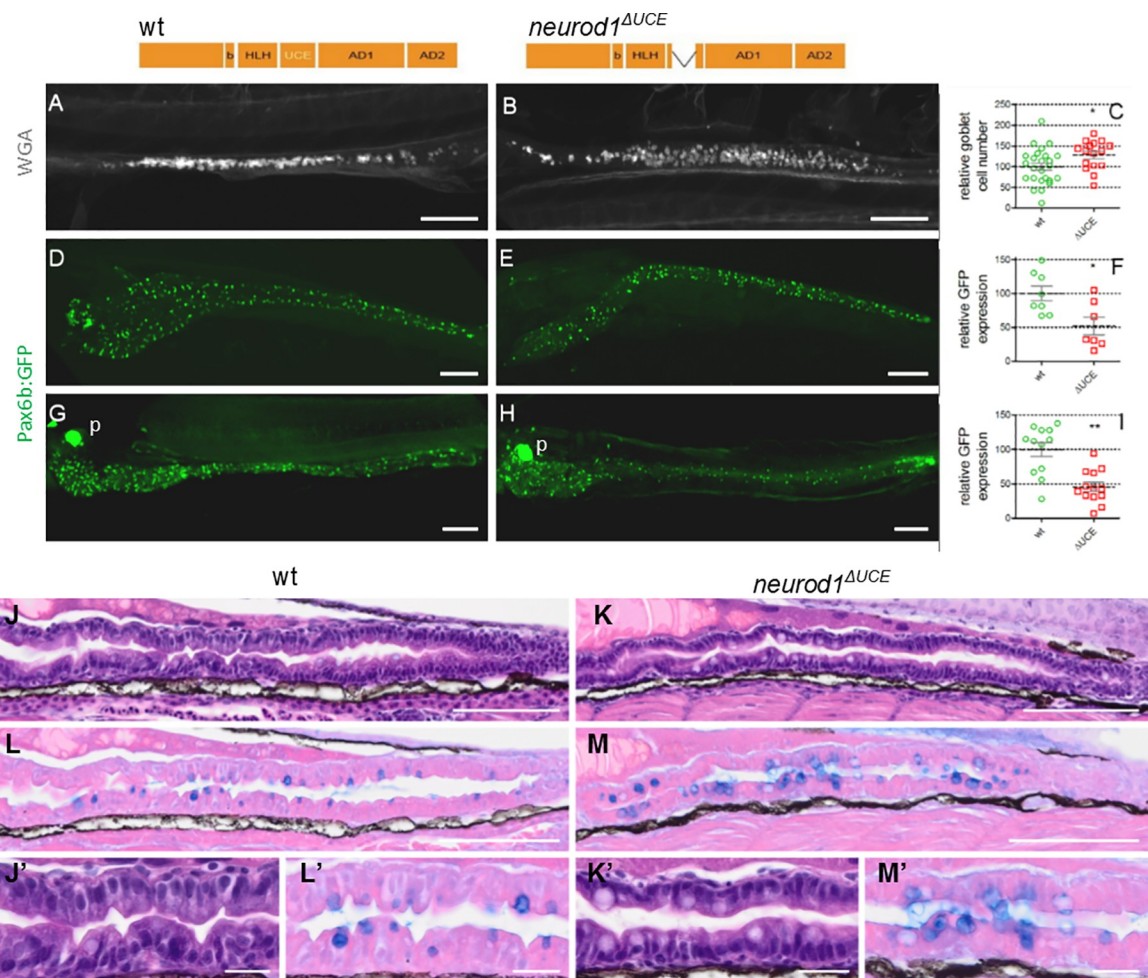

**Fig 8. *neurod1^ΔUCE* homozygous mutants display an increase of the goblet cells concomitant with a decrease of EECs. Top:** Schematic representation of Neurod1 and of Neurod1^ΔUCE mutant, harbouring an 81-bp deletion spanning the 19aa of the UCE and 3 and 5 aa upstream and downstream of the UCE, respectively. **A-C:** Immunodetection of goblet cells labelled with a rhodamine dextran-conjugated wheat germ agglutinin (WGA) that interacts with the N-acetylglucosamine of the mucus. Quantification was done by counting the goblet cells under a fluorescent stereomicroscope. **D-I:** Immunodetection of GFP performed on *pax6b:GFP* transgenic embryos at 5 dpf (D-F) or 4 dpf (G-I). The location of the pancreas (p) is indicated on G and H. Quantification of the relative GFP expression was performed by quantifying the volume occupied by the GFP cells using the Imaris software. Asterisks indicate that the difference between the cell number in wt controls and *neurod1^ΔUCE* mutants is statistically significant using the Mann-Whitney U-test (**: P <0.01; *: P <0.05). Views are lateral (A-B) or ventral (D-H) with the anterior part to the left and represent confocal projection images. Scale bars: 100μm. **J-M:** Subsequent 5 μm microtome sections of 5 dpf wt (J, L) or Neurod1^ΔUCE mutant (K,M) larvae stained with haematoxylin/eosin (J-K) or with eosin/alcian blue (L-M). Scale Bars: 100 μm. J'-M': enlarged view of J-M: Scale Bars: 20 μm.

expression level >10 CPM) (305 up- and 375 down-regulated genes) (S3 Table). Concerning the expression of EEC hormones, we observed a drastic decrease of *tac1*, *ghrl*, *mlnl* and *nmbb*, a 2 to 3 X reduction of *gcga*, *calca* and *adcyap1a* and a downward trend for *insl5a* and *galn* (Fig 9A1). WISH analyses confirmed a reduction in the number of cells expressing *mlnl* (4.4X***), *adcyap1a* (2.0X*), *insl5a* (1.5X**,) *gcga* (2.3X***) and *galn* (3.6X*) in the mutant (Fig 9B and 9A2)). The RNA-seq analyses detected a higher *penka* expression in *neurod1^ΔUCE* compared to wild-type larvae (2.7-fold); however this result was not confirmed by IHC (Fig 9B and 9A2). This apparent discrepancy could be due to the 2-fold reduction in the number of *pax6b:GFP^+* cells (Fig 8G–8I). Indeed, an unchanged number of *penka^+* cells in the *neurod1^ΔUCE* intestine

A

| Name | Neurod1$^{ΔUCE}$ | | | | | | Neurod1$^{-/-}$ | |
| | A1 : RNAseq | | | | A2 : FISH or IHC | | A3 : FISH | |
| | Mean WT | Mean neu-rod1$^{ΔUCE}$ | Log2 FC | FDR | Log2 FC | FDR | Log2 FC | FDR |
|---|---|---|---|---|---|---|---|---|
| tac1 | 39 | 1 | -5,07 | 0,0000 | | | | |
| ghrl | 142 | 6 | -4,68 | 0,0000 | | | | |
| mlnl | 592 | 61 | -3,48 | 0,0000 | -2,19 | <0.0001 | -0,23 | 0,9230 |
| nmbb | 1059 | 119 | -3,37 | 0,0000 | | | -5,29 | <0.0001 |
| gcgb | 98 | 26 | -1,94 | 0,7322 | | | | |
| gcga | 1431 | 450 | -1,81 | 0,0855 | -1,18 | 0,0074 | -0,03 | 0,3795 |
| insl5b | 135 | 45 | -1,81 | 0,1157 | | | | |
| galn | 658 | 227 | -1,75 | 0,1958 | -1,47 | 0,0401 | -1,32 | <0.0001 |
| insl5a | 3643 | 1717 | -1,32 | 0,2039 | -0,63 | 0,0038 | -0,59 | 0,0061 |
| calca | 1706 | 911 | -1,06 | 0,0165 | | | -2,07 | <0.0001 |
| adcyap1a | 6140 | 3270 | -1,05 | 0,0308 | -0,97 | 0,0401 | -0,98 | 0,0018 |
| pyyb | 31190 | 21163 | -0,72 | 0,4476 | -0,33 | 0,0550 | -0,32 | 0,1073 |
| pyya | 222 | 226 | -0,16 | 0,8866 | | | | |
| ccka | 5509 | 5711 | -0,11 | 0,9511 | | | -0,77 | 0,0061 |
| vipb | 496 | 569 | -0,09 | 0,9714 | | | | |
| penka | 13211 | 34016 | 1,32 | 0,0930 | -0,13 | 0,7106 | 0,32 | 0,9648 |
| sst1.2 | 45 | 220 | 2,19 | 0,6148 | | | | |
| sst2 | 70 | 363 | 2,29 | 0,1458 | | | | |
| gip | 314 | 2882 | 3,19 | 0,0813 | | | | |
| sst1.1 | 13 | 716 | 5,34 | 0,2383 | | | | |

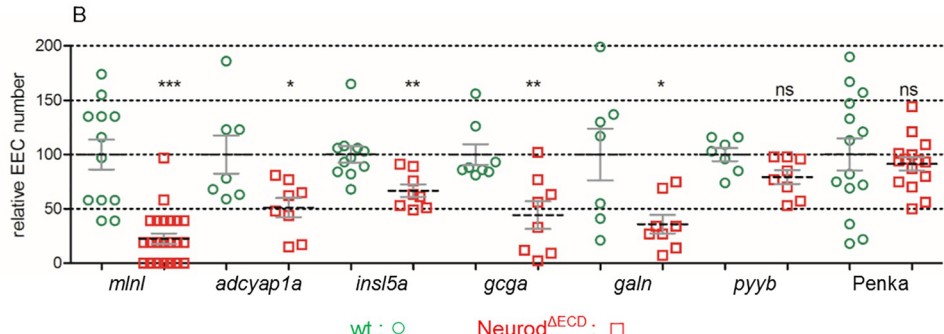

B

**Fig 9. *neurod1$^{ΔUCE}$* homozygous mutants display a decrease in the expression of several EE hormones. A1:** The expression level (given in normalized CPM) of the EE hormones was obtained from the RNA-seq data of *Tg(pax6b: GFP$^+$)* EEC isolated by FACS from wt or *neurod1$^{ΔUCE}$* homozygous mutant larvae at 4 dpf. The values are the expression mean of 6 wt and 4 *neurod1$^{ΔUCE}$* samples. Genes underlined in grey are not significantly regulated (FDR > 0.1). S3 Table provides the expression level of all differentially expressed genes (values for each sample, means, log2 FC and FDR). **A2.** Log2 fold change (FC) and FDR in the hormone cell number observed in wt compared to *neurod1$^{ΔUCE}$* homozygous mutant embryos as quantified on Fig 9B. **A3.** Log2 FC and FDR in the hormone cell number observed in control siblings compared to *neurod1$^{-/-}$* homozygous mutant embryos as shown on Fig 11C. Grey zones indicate non-significant change: FDR >0.05). **B:** Quantification of the number of EECs detected either by FISH performed at 96hpf with *mlnl, adcyap1a, insl5a, gcga, galn and pyyb* probes or by immunodetection of Penka cells on wt or *neurod1$^{ΔUCE}$* homozygous mutant embryos. Quantification was done by counting the cells using the Imaris software after confocal scanning except for *mlnl* were the cells were directly counted under a fluorescent stereomicroscope. The mean is indicated by a dashed line and the S.E by solid lines. Asterisks indicate that the difference between the cell number in wt controls and *neurod1$^{ΔUCE}$* mutants is statistically significant using the Mann-Whitney U-test (***: P <0.001; **: P <0.01; *: P <0.05, ns: P >0.05).

will result into a 2-fold increase in the proportion of *penka+* cells among the remaining *pax6b: GFP$^+$* sorted cells.

Most of the transcription factors known to be important for endocrine cell differentiation [16] are unchanged except *sox4b* and *pax4* which were respectively down- and up-regulated in *neurod1$^{ΔUCE}$* (Fig 10A). Analysis of all DE genes highlighted *cuzd1.1* and *cuzd1.2* genes as highly

| | Name | Mean WT | Mean neurod1$^{\Delta UCE}$ | Log2 FC | Padj |
|---|---|---|---|---|---|
| A | *pax4* | 269 | 578 | 1.03 | 0.0171 |
| | *rfx6* | 90 | 124 | 0.50 | 0.8080 |
| | *isl1* | 185 | 256 | 0.43 | 0.4476 |
| | *nkx2.2a* | 220 | 301 | 0.35 | 0.5740 |
| | *fev* | 944 | 1210 | 0.25 | 0.6620 |
| | *neurod1* | 669 | 820 | 0.21 | 0.8043 |
| | *pdx1* | 510 | 593 | 0.10 | 0.8678 |
| | *insm1a* | 141 | 149 | -0.02 | 0.9783 |
| | *insm1b* | 706 | 694 | -0.11 | 0.9331 |
| | *ascl1a* | 26 | 24 | -0.18 | 0.9359 |
| | *pax6b* | 591 | 513 | -0.35 | 0.6470 |
| | *sox4b* | 30 | 7 | -2.26 | 0.0006 |
| | | | | | |
| B | *cuzd1.1* | 15 | 211 | 3.7 | 0.09334 |
| | *cuzd1.2* | 7 | 353 | 5.5 | 0.00515 |
| | *mfsd4aa* | 3 | 107 | 5.2 | 0.00000 |

**Fig 10. Expression level of selected transcription factors in EEC from wild-type or *neurod1*$^{\Delta UCE}$ homozygous mutant larvae. A:** Expression level for 12 "endocrine pancreas development" genes. **B:** Expression level of selected genes differentially up-regulated by *neurod1*$^{\Delta UCE}$. The expression level (given in normalized CPM) was obtained from the RNA-seq data of *pax6b*:GFP+ EEC isolated by FACS from wt or *neurod1*$^{\Delta UCE}$ homozygous mutant larvae at 4 dpf. The values are the expression mean of 6 wt and 4 *neurod1*$^{\Delta UCE}$ replicates. Genes underlined in grey are not significantly regulated (FDR > 0.1). S3 Table provides the expression level of all differentially expressed genes (values for each sample, means, log2 FC and FDR).

upregulated in *neurod1*$^{\Delta UCE}$ (Fig 10B and S3 Table). These two proteins are orthologous to the mammalian CUZD1, known to be a critical mediator of the JAK/STAT5 signaling pathway [23], important for intestinal homeostasis [24]. A strong upregulation of *mfsd4aa* (major facilitator superfamily domain-containing protein 4) (~18X) is also observed (Fig 10B) which was also specifically induced by the Tg7 (ΔUCE) and Tg8 (ΔECD) GOF (Fig 6C, panel 3). MFSD4AA is a solute carrier protein (SCL) putatively involved in the transport of growth factors [25]. A Gene Ontology analysis performed on these DE genes shows a significant enrichment for insulin signaling pathway (p adjust = 1.2 E-2) with 13 out of 60 genes of the cascade being differentially expressed (S5 Fig).

## Neurod1$^{-/-}$ displays an increase of goblet cells concomitant with a reduction of EECs like Neurod1$^{\Delta UCE}$

Finally, to further investigate the requirement of the UCE domain for Neurod1 activity, we compared the phenotype of the *neurod1*$^{\Delta UCE}$ with a null *neurod1* mutant. The zebrafish mutant line *neurod1*$^{ulg007}$ was generated by CRISPR/Cas9 mutagenesis and contains a 4 bp deletion in the coding region of Neurod1. This leads to a frameshift in the basic domain at aa 102 leading to a short aberrant 36-aa ORF and an early termination codon. Immunohistochemistry (IHC) performed on 4 dpf *neurod1*$^{ulg007/ulg007}$ larvae revealed a complete loss of Gcg and a significant reduction of Sst in the pancreas (Fig 11A), confirming our previous results obtained at 30 hpf using morpholinos [8]. This strongly suggests that the *neurod1*$^{ulg007/ulg007}$ mutant represents a bona fide null mutant, called hereafter *neurod1*$^{-/-}$.

We next analysed the goblet and EECs in the intestine of *neurod1*$^{-/-}$ larvae. We found a significant increase in the number of goblet cells detected in the intestine of *neurod1*$^{-/-}$ compared to control sibling embryos at 4 dpf and a concomitant decrease of *pax6b*+ EECs (Fig 11B), like observed in *neurod1*$^{\Delta UCE}$ mutant. Further analyses of the EECs show that, like in *neurod1*$^{\Delta UCE}$, we observed a large decrease in the number of *nmbb*+ cells (39-fold) and a similar reduction in the number of cells expressing *galn* (2.5 fold), *adcyap1a* (2.0 fold), *calca* (4.2 fold) and *insl5a* (1.5 fold) (Fig 11C and see the comparison with *neurod1*$^{\Delta UCE}$ in Fig 9A3). Furthermore, like

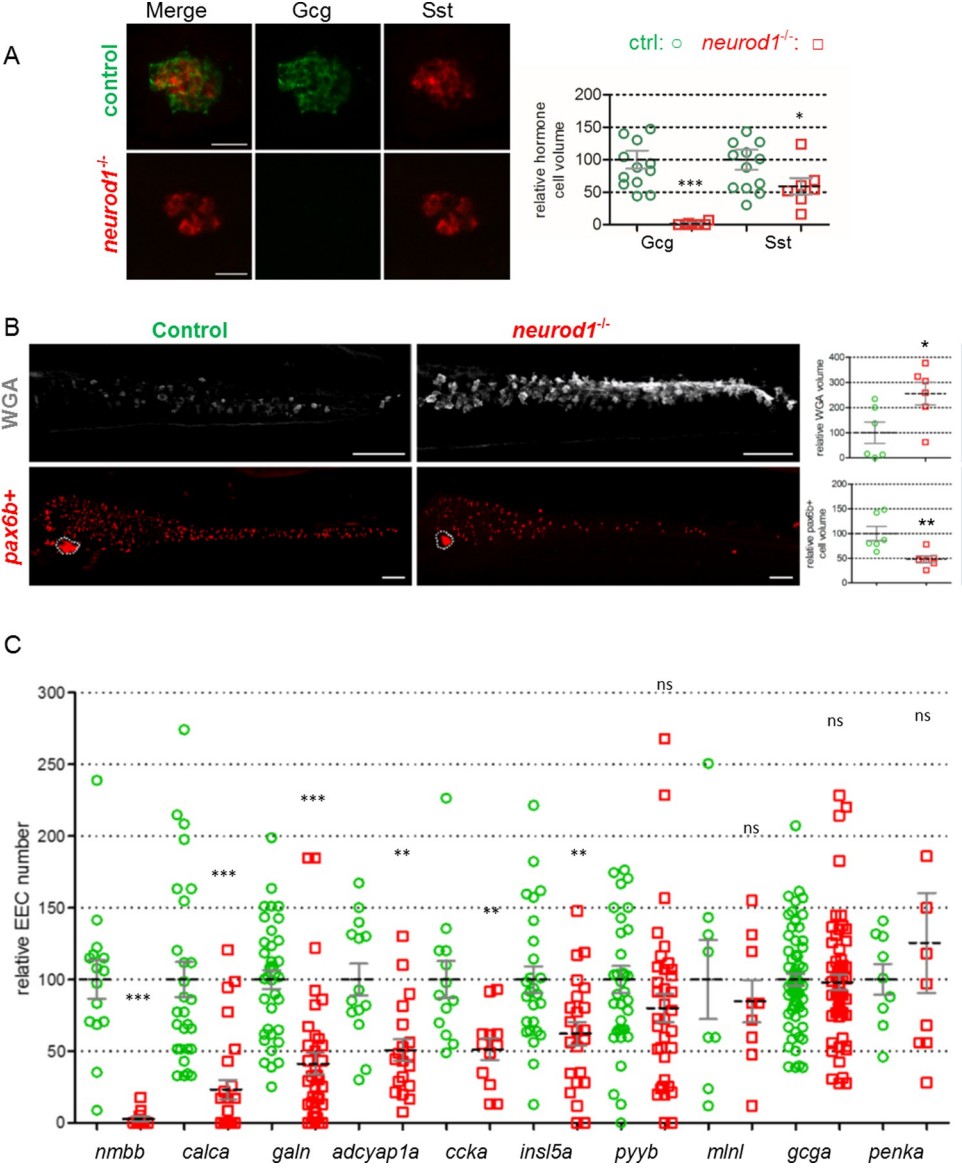

**Fig 11. neurod1⁻/⁻ displays an increase of goblet cells concomitant with a reduction of EECs. A:** Immunodetection at 96hpf of Gcg and Sst cells of *neurod1⁻/⁻* homozygous mutant compared to control sibling embryos. Quantification was performed by quantifying the volume occupied by the cells using the Imaris software. Views represent confocal projection images. Scale bars: 20 μm. **B:** Immunodetection at 96hpf of goblet cells (WGA) or *pax6b+* cells of *neurod1^ΔUCE* homozygous mutant compared to control sibling embryos. Quantification was performed by quantifying the volume occupied by the cells using the Imaris software. Views represent confocal projection images with the anterior part to the left. The pancreas are encircled with dotted lines. Scale bars: 50 μm. **C:** Quantification of the relative number of EECs detected by FISH performed at 96hpf on *neurod1^ΔUCE* homozygous mutant compared to control sibling embryos. Quantification was done by counting the cells under a fluorescent stereomicroscope. The mean is indicated by a dashed line and the S.E by solid lines. Asterisks indicate that the differences between controls and *neurod1⁻/⁻* mutants are statistically significant using the Mann-Whitney U-test (***: P <0.001; **: P <0.01; *: P <0.05, ns: P >0.05).

*neurod1^ΔUCE*, we did not observe any change in the number of *pyyb* and *penka* cells (Figs 11C and 9A3). Only three of the tested hormones are expressed differentially in *neurod1⁻/⁻* and *neurod1^ΔUCE* mutants: the expression of *gcga* and *mlnl* are unchanged in *neurod1-/-* while these 2

hormones were reduced in *neurod1*<sup>ΔUCE</sup> and *ccka* was reduced in *neurod1-/-* (1.7X) while it is not in *neurod1*<sup>ΔUCE</sup> (Figs 11C and 9A).

In conclusion, *neurod1-/-* and *neurod1*<sup>ΔUCE</sup>mutants both show an increase of goblet cells and a general reduction of EECs. Furthermore, similar subsets of EEC are affected as 7 out of the 10 EEC analysed present analogous perturbations. Altogether these data support the crucial function of the UCE domain for NeuroD1 activity in the intestine.

## Discussion

ARP/ASCL factors are key determinants of cell fate specification in a wide variety of tissues, coordinating the acquisition of generic cell fates as well as of specific subtype identities. We have previously shown that the identity of these determinants for a given cell fate is not always conserved throughout vertebrate evolution suggesting functional equivalence between these factors [5,8]. To determine the common and specific properties of ARP/ASCL factors, we performed GOF experiments to define which ARP/ASCL factors are able to rescue the secretory cells in the intestine of the zebrafish *ascl1a-/-* mutant. Atoh1a/b, Ascl1a/b and Neurod1 were all able to initiate the secretory cascade in the *ascl1a-/-* mutant as shown by the activation of *sox4b*. However, Atoh1a/b and Ascl1a/b rescued only the goblet cell lineage while Neurod1 rescued exclusively the EECs. Functional analysis of several Ascl1a/Neurod1 chimeric proteins allowed us to identify a domain in Neurod1 essential for its function. This domain is highly conserved in all Neurod proteins and absent in the other ARP/ASCL proteins and was called here UCE for ultra-conserved element. The UCE domain is required for the optimal transactivation properties of Neurod1 but also for repressing the expression of the Gfi1aa factor, known to be an important factor in goblet cell differentiation in mouse. Deletion of the UCE domain in the endogenous Neurod1 protein further underlines its importance as the *neurod1*<sup>ΔUCE</sup> mutant displays an increase in the number of goblet cells concomitant with a reduction of several EE subtypes. Importantly, the *neurod1* null mutant displays very similar defects supporting the crucial function of the UCE domain for NeuroD1 activity in the intestine. Our findings therefore unveil common as well intrinsic specificities within ARP/ASCL factors.

We have shown here that the capacity to induce the secretory cascade is shared by Ascl1, Atoh1 and Neurod1, which is reminiscent to the capacity of all proneural factors to induce the neuronal program. Furthermore, like in the nervous system where the subtype specification depends on specific factors, we found here that only Neurod1 is able to induce the enteroendocrine program. This is due to its capacity to induce several genes important for endocrine differentiation such as *pax4*, *rfx6*, *nkx2.2a* and *insm1a/b* while these endocrine genes are not activated at all by Ascl1a (Fig 6C, panel 1). This unique property of Neurod1, that cannot be fulfilled by the other ARP/ASCL factors, explains the omnipresence of Neurod1 in all endocrine cells of the intestine but also of the pancreas in zebrafish and, to our knowledge, this is also the case in most vertebrates. This contrasts with the flexibility observed notably in the choice of the secretory cell fate determinants in vertebrates, made possible given the shared capacity of inducing the secretory lineage by several ARP/ASCL factors as shown here.

Besides these endocrine genes, Neurod1 activate also two important actors for Retinoic acid signalling (*crabp1* and *greb1l*) while these two genes are repressed by Ascl1a. CRABP1 (cellular retinoic acid binding protein 1) binds to retinoid acid (RA) helping its transport into the nucleus where it will bind its receptor (RAR) and activates retinoid acid-dependent genes. GREB1L resides in a chromatin complex with RAR members where it acts as a coactivator for RARs [26]. Although the influence of retinoic acid signalling on EEC fate is not known, in the pancreas, retinoic acid promotes the generation of pancreatic endocrine progenitor cells and their further differentiation into beta-cells [27,28]. As pancreatic and intestinal endocrine cells

share transcriptomic signatures and regulatory programmes [16], we can postulate that RA could have also a positive effect on endocrine cell differentiation in the gut. The same inverse regulation by Neurod1 and Ascl1a is also observed for an important actor of EEC differentiation, *runx1t1*, activated by Neurod1 and repressed by Ascl1a. RUNX1T1 has been recently shown to be transiently activated in EE progenitors and its inactivation affect *GIP*, *Cck* and *Sst* expression in intestinal organoids [18].

Inverse regulation by Ascl1a and Neurod1 is also found for the ligands of the Delta/Notch signalling pathway. Indeed we found that Ascl1a overexpression represses the expression of *deltaB (dlb)* and *delta-like 4 (dll4)* while Neurod1, Neurod1$^{\Delta UCE}$ and Neurod1$^{\Delta ECD}$ activate *dlb*, *dlc* and *dll4* (Fig 6C, panel 2). The opposite regulation by Ascl1a and Neurod1 of these Notch ligands suggests an inverse influence on the Delta/Notch signalling that could influence goblet versus EEC fate decision. Notch signalling has been extensively described to be involved in the absorptive versus secretory cell fate choice [29] but not in the subsequent step leading to either EECs or paneth/goblet cells. Further research will be required to resolve this question.

## The UCE, rather than the bHLH domain, encodes important features for Neurod1 specificity

We demonstrate in the present study that the basic and HLH domains of Neurod1 can be exchanged with those of Ascl1a without affecting the induction of the EEC. This contrasts with the results described for Atoh and Neurog factors where the functional divergence is encoded by three residues in the basic domain [30]. We identify an evolutionary conserved domain with a central element (UCE) nearly identical in all Neurod members that is a repressor of the goblet lineage. RNA-seq data show that the UCE domain acts by repressing the expression of the zinc finger factor Gfi1aa in goblet cells. In mouse, *Gfi1* is expressed in goblet and Paneth cells but not in the EEC lineage [31]. In the murine *Gfi1*-deficient intestinal epithelium, goblet and Paneth cell lineages aberrantly express the pro-enteroendocrine transcription factor *Neurog3* and consequently undergo reprogramming into the EEC lineage [31]. This indicates that, in the intestinal epithelium, *Gfi1* helps to stabilize the goblet and Paneth cell lineages by participating to the repression of *Neurog3*. Such role of GFI1 (and of Senseless, its orthologous gene in drosophila) as a binary switch favouring one lineage over another has been described in many tissues. For instance, GFI1 controls sensory organ precursor selection [32] or colour photoreceptor differentiation in Drosophila [33]. In murine hematopoiesis, GFI1 acts as a transcriptional repressor that recruits the histone demethylase complex LSD1/CoREST to epigenetically silence the endothelial program in hemogenic endothelium and allows the emergence of blood cells [34]. In the inner ear, this binary switch involves Neurod1. Indeed GFI1 promotes hair cell development by repressing neuronal-associated genes such as Neurod1 and by activating hair cell-specific genes required for cell maturation [35]. Based on our study, we can postulate that, in the inner ear also, NEUROD1 will repress the expression of *Gfi1* through its UCE domain creating a mutual cross-repression between NEUROD1 and GFI1. Such antagonism could also occur in the pancreas where GFI1 has been shown to be involved in exocrine cell differentiation [36] while NEUROD1 is known to be important for endocrine cell differentiation [37], these two cell types originating from the same precursor.

What is the mode of action of this UCE? Its position between the HLH and the transactivation domain of Neurod1 suggests that it does not directly interact with DNA but could interact with co-factors. Unfortunately, we did not succeed to identify interacting partners to this region using high-throughput yeast-two hybrid (YTH) screenings. Nevertheless, examination of the UCE protein sequence reveals a phosphorylation consensus sequence for the ATM/ATR/DNA-PK protein kinases (S/TQ) (residue S172 in dr-Neurod1). Interestingly, this site

lies at the centre of a longer cluster containing two other S/TQ motifs (TQ at aa129 and SQ at aa195) and therefore constitutes a so-called SCD domain, which is defined by the presence of at least three S/TQ sites in a stretch of 100 amino acids [38]. More than half of the characterized ATM targets contain an SCD [38]. Although ATM/ATR are cellular kinases with a well-characterized role in DNA-damage response, there is growing evidence that they also act in other pathways and notably in the insulin-signalling pathway [39] for which we identify many DE genes in $neurod^{\Delta UCE}$ mutant larvae. Furthermore, the SCD domain is over-represented in proteins involved in the nervous system development and is present for example in EMX2, PAX6, GLI2 and GLI3 [38]. All these data suggest a much broader role of these kinases than initially thought. Further studies will be necessary to determine the exact role of these ATM/ATR sites in Neurod1. But it became clear recently that phosphorylation constitutes an important mechanism of regulation of ARP/Ascl activity that could affect their specificity. For instance, mutating Neurod1 at the glycogen synthase kinase 3β consensus site ($S^{274}$A in Xneurod) changes the timing of XNeuroD function, so that it can promote the differentiation of early retinal cell types while wild-type XNeuroD can induce only late retinal cell types [40].

## The $neurod^{\Delta UCE}$ and the $neurod$-/- mutant display an increase of the goblet cells concomitant to a decrease of EEC

An increase in the number of goblet cells was detected in $neurod^{\Delta UCE}$ mutants consistent with the GOF experiments indicating that this domain functions as a repressor of the goblet cell fate. We also observed a concomitant decrease in the number of EECs which could be a consequence of the increase of the goblet cells (secretory cells choosing one fate or another). The $neurod1^{-/-}$ mutant displays a very similar intestinal phenotype as $neurod1^{\Delta UCE,}$ both showing an increase of goblet cells and a general reduction of EEC. These data indicate that the ECD is crucial for the activity of Neurod1 in the intestine. However it is probably not the case for all tissues as the $neurod1^{-/-}$ homozygous mutant can not reach the adulthood in contrast to $neurod^{\Delta UCE.}$ It will be therefore interesting to compare in the future the phenotype of these two mutants in different tissues like the central nervous system, the pancreas and the inner ear.

## Methods

### Ethics statement

All animal experiments were conducted according to national guidelines and were approved by the ethical committee of the University of Liège (protocol numbers 13–1557 and 19–2085.

### Zebrafish maintenance, transgenic and mutant lines

Zebrafish (Danio *rerio*) were raised according to standard protocols and staged according to Kimmel [41]. The *pia* mutant (*pia$^{t25215}$* mutant allele of *ascl1a*) was provided by M. Hammerschmidt and genotyped as described [42] or by in situ using a prolactin (prl) probe as the *ascl1a*-/- loses pituitary *prl* expression. The following zebrafish transgenic lines were used: *Tg (pax6b:GFP)$^{ulg515}$* [22], *Tg(hsp70:atoh1a)*, kindly provided by Bruce Riley [15], the *Tg(hsp70: atoh1b-Myc)*, kindly provided by Sudipto Roy [14] and the Tg(sox17:dsred), kindly provided by Stephanie Woo [43].

### Generation of the *Tg(hsp70l:eGFP-2A-TF)* lines

The *Tg(hsp70l:eGFP-2A-Ascl1a) (Tg1)*, where the heat-shock inducible promoter *hsp70l* controls the expression of a bicistronic transcript, eGFP-2A-Ascl1a, allowing the production in

equimolar quantities of eGFP and Ascl1a from a single transcript [44], were generated by constructing first the bicistronic *sequence eGFP-2A-Ascl1a* by an overlap extension polymerase chain reaction (OE-PCR). The OE-PCR method consists of two primary PCR reactions which generate DNA fragments with overlapping ends (see S4 Table for the details on the primers and the PCR) and a secondary reaction which joins the two fragments into a single one [45]. The resulting fragment *eGFP-2A-Ascl1a* were then cloned in the PCR8-GW vector in the sense orientation. Then, a triple LR recombination was performed with 5E-HSP (Tol2 kit #222 [46]), pCR8-GW- *eGFP-2A-Ascl1a* and p3E-polyA (Tol2kit #302) and the vector pDestTol2CG2 (Tol2 kit #393) to generate the transgene Tg*(hsp70l:eGFP-2A-Ascl1a)*. This transgene has been then introduced into AB embryos by co-injection with the Tol2 transposase to generate the Tg *(hsp70l:eGFP-2A-Ascl1a)* [ulg056Tg] line. Similar protocol was used for the transgenes Tg2 to Tg7 using the primers listed in the S4 Table. The transgene Tg8 that contains a deletion of the ECD was performed by Q5 Site-directed mutagenesis (NEB) using the 2 primers listed in the S4 Table.

Heat shocks were performed by incubation of the larvae at 39˚C during 30 minutes.

## Isolation of EECs and endodermal cells by FACS

EECs were isolated by dissecting the gut from about 200 transgenic neurod1[ΔUCE/ΔUCE] or wild-type *Tg(pax6b:GFP)*[ulg515] larvae at 4 dpf, taking care of not including pancreatic tissue. Cell dissociation was performed by incubation in HBSS 1x supplemented with 100 U/ml collagenase IV and 0.3 U/ml Dispase (Life Technologies) for 10 minutes. Cells were washed in HBSS ($Mg^{2+}$ and $Ca^{2+}$ free) containing 1% BSA and GFP-expressing EECs were selected by FACS purification using FACS Aria II. This procedure allows us to obtain between 1000 to 2000 isolated GFP+ cells.

Endodermal *sox17-dsred+* cells were isolated by dissecting the trunk from about 150 double transgenic *Tg(hsp70l:eGFP-2A-ARP/Ascl)* or wild-type *Tg(sox17:dsred)* larvae at 52 dpf, heat-shocked at 38 and 48hpf, taking care of not including pancreatic tissue. Cell dissociation was performed by incubation in TrypLE select 1X (Gibco) for 8 minutes. Cells were washed in HBSS (Mg2+ and Ca2+ free) containing 1% BSA and *dsred*[+] EECs were selected by FACS purifications using FACS Aria II. This procedure allow us to obtain between 2000 to 4000 isolated *dsred+* cells.

## cDNA synthesis, library preparation and sequencing

Each FACS-sorted cells sample was directly pelleted by centrifugation and resuspended in 3.5 μl of *reaction buffer*, lysed by freezing in liquid nitrogen and stored at -80˚C according the Smart-seq2 protocol [47]. cDNA was synthesised and amplified by a 13 cycles PCR reaction. Quality of cDNA was verified by 2100 High Sensitivity DNA assay (Agilent technologies). 1 ng cDNA was used for preparing each cDNA library using Nextera-XT kit (Illumina) and sequenced on Hi-seq 2000 to obtain around 20 millions of reads (75 base single-end).

## RNA-seq data analyses

Sequences were trimmed in order to remove adaptors and low quality bases. Trimmed reads were mapped in to the zebrafish genome GRCz11 (Ensembl Release 92, www.ensembl.org) using STAR software v.2.5.4b [48]. Gene expression was measured from the mapped reads by using built-in STAR module (—quantMode GeneCounts) and are expressed in counts of reads per million (CPM) [49]. The comparison of the transcriptomes as well as differential expression (DE) analysis were performed using DESeq2 R package [50] to identify all genes displaying significant change in expression between two conditions (with Fold-Discovery-

Rate < 10%). For the analysis of the GOF RNAseq data, we constructed *in silico* one gene corresponding to the region of Neurod1 used in the transgene (ENSDARG00000139139). To quantify the transcripts corresponding only to the 3' UTR of *neurod1*, not contaminated by the transcripts originating from the transgene, all the reads mapping to both the transgene (ENSDARG00000139139) and neurod1 (ENSDARG00000019566) were not taking into account. For *ascl1a*, the Ensembl ID ENSDARG00000038386 corresponds only to 3'UTR of the endogenous *ascl1a* gene whereas the annotation ENSDARG00000101628 (erroneously annotated ascl1b) correspond actually to the ORF sequence of ascl1a. The RNAseq data have been deposited in Gene Expression Omnibus (GEO) with the accession number GSE193281 and 3 additional wt control at 4 dpf were provided by [16] (GSE149081)

## Gene Ontology enrichment analysis

Gene Ontology (GO) enrichment analysis was performed on the different gene sets using the PANTHER bioinformatics resources 16.0 [51] taking as background all the zebrafish genes. The enrichment analysis was focused on the GO biological process, molecular function and KEGG pathways with a statistical Fisher exact test p-value<0.05.

## Riboprobes, whole-mount in situ hybridization (WISH) and Immunohistochemistry

Antisense riboprobes were made by transcribing either linearized cDNA clones as described in [52] or T3-flagged (CCCAATTAACCCTCACTAAAGGGAG) PCR probes as described in [53]. The zebrafish agr2 [54], sox4b [55], pax6b [56], somatostatin2 [57] and gcga [58] probes have been described elsewhere. *pyyb* probe was obtained by transcribing with Sp6 the Sal1-digested pSPORT1-pyyb plasmid (accession number BF17422). T3-flagged probes were generated for mlnl, galn, adcyap1a, insl5a, gfi1aa and for the 3'UTR neurod1 probe, using the primers listed in S4 Table. Wholemount in situ hybridization (WISH) and fluorescent in situ hybridization (FISH) were performed as described previously [55].

Immunohistochemistry (IHC) on whole-mount embryos/larvae was performed as described [8]. The antibodies used are the chicken anti-GFP (Aves lab) used at a dilution of 1000 X, the rabbit anti-Enkephalin (T4294; Peninsula Laboratories International, Inc) used at a 400X dilution, and Alexa Fluor secondary antibodies at 1000X dilution. Goblet cell mucin was detected with rhodamine conjugated wheat germ agglutinin (1,100 dilution) (Vector Laboratories). Stained embryos were mounted in Prolong (Invitrogen) with DAPI and imaged using SP5 confocal microscope (Leica). Quantification was performed either by counting the cells under a fluorescent microscope or either by determining the volume occupied by the cells using the Imaris software (Bitplane). Haematoxylin/eosin and eosin/alcian staining was performed on 5µm microtome sections by the GIGA immunohistology platform and the sections were recorded using the SLIDEVIEW VS200 slide scanner Olympus. Statistical analysis was performed using the non-parametric Mann-Whitney t-test of the GraphPad Prism software. Creation of the *figures* was achieved using ImageJ and its plugin FigureJ [59]

## CRISPR/cas9 genome mutagenesis

The neurod1^ulg007 and neurod1^ulg052 mutant lines, called here respectively neurod1-/- and neurod1^ΔUCE, were generated by CRISPR/Cas9. The first one was generated by cloning the annealed BP720 and BP721 oligos (S4 Table) into the BsaI-digested DR274 plasmid (Addgene) that was transcribed in gRNA as described [60]. The Cas9 mRNA were transcribed using PmeI-digested Cas9 expression vector NLM3613 (Addgene) as described [60]. Zebrafish eggs

were injected with ~1 nL of a danieau solution containing ~60ng/μl of the gRNA and ~300 ng/μl of Cas9 mRNA. For the second line. CRISPR guide RNAs were selected using chopchop software (http://chopchop.cbu.uib.no/) to target flanking regions of the UCE. The cloning-free single-guide RNA (sgRNA) synthesis was done as described [61] by annealing the sgRNA core oligo O490 with O671 or with O672. The Streptococcus Pyogenes cas9-NLS-GFP protein was produced from the pMJ922 expression plasmid (Addgene), purified and cleaved as described in [62]. After synthesis and purification of gRNA, fertilized zebrafish eggs were injected with approximately 1 nL of a danieau solution containing 120ng/μl of the gRNA, 760 ng/μl of purified Cas9 protein.

The efficiency of mutagenesis was verified by genotyping using Heteroduplex Migration Assays [63] after amplification of targeted genomic sequences. Founder fish transmitting germline mutations were outcrossed with wild type fish.

## Supporting information

**S1 Fig. Rescue of *ascl1a-/-* by Ascl1b and Atoh1a. A–C**: FISH performed at 55 hpf with a *sox4b* probe on ascl1a<sup>-/-</sup> or control sibling embryos heat-shocked at 36 and 46 hpf. **D-I**: FISH performed at 96 hpf with the *agr2* probe (D-F) and a mix of hormone probes (*ghrelin (ghrl)*, *peptide YYb (pyyb)*, *glucagon-a (gcga)* and *somatostatin-2 (sst2)) (G-I)* on ascl1a<sup>-/-</sup> or control sibling embryos heat-shocked at 36, 46 and 56 hpf. The transgenic line used is indicated on the left part of the figure as well as the genotype of the larvae; the *ascl1a*<sup>-/-</sup> larvae were identified by the loss of the pituitary *prl* expression (not shown). The pancreas is encircled while the location of gut, visualised with a DAPI staining (not shown), is delimited by dashed lines. All Views are ventral with the anterior part to the left. Scale bar: 100 μm.
(TIF)

**S2 Fig. Neurod1 is able to rescue *adcyap1a*<sup>+</sup>, *gcga*<sup>+</sup>, *insl5a*<sup>+</sup> and *pyyb*<sup>+</sup> EEC of *ascl1a-/-* larvae. A-D:** FISH performed at 96hpf with *adcyap1a (A)*, *gcga (B)*, *insl5a (C) and pyyb (D)* probes on ascl1a-/-; Tg(hsp70l:eGFP-2A-Neurod1) larvae heat-shocked at 36, 46 and 56hpf. The ascl1a-/- larvae (not shown) do not express any hormones [5]. All views acquired with a fluorescence microscope are ventral with the anterior part to the left. The pancreas is encircled while the location of gut, visualised with a DAPI staining (not shown), is delimited by dashed lines. Scale bar: 100μm.
(TIF)

**S3 Fig. Atoh1b and Ascl1a are not able to rescue the expression of Neurod1 in *ascl1a-/-* larvae.** FISH performed at 72hpf with a 3'UTR *neurod1* probe on wild-type (A), *ascl1a*<sup>-/-</sup> (B), *ascl1a*<sup>-/-</sup>; Tg(hsp70l:atoh1b-Myc) (C) ascl1a<sup>-/-</sup>; Tg(hsp70l:eGFP-2A-ascl1a) (D) and ascl1a<sup>-/-</sup>; Tg (hsp70l:eGFP-2A-neurod1) (E) larvae, heat-shocked at 36, 46 and 56hpf. All views are ventral with the anterior part to the left and represent confocal projection images. The pancreas is encircled while the location of gut, visualised with a DAPI staining (not shown), is delimited by dashed lines. Scale bar: 100μm.
(TIF)

**S4 Fig. Analysis of EEC inducing capacities of Ascl1a, Neurod1 and the chimeric Ascl1a/Neurod1 proteins. A-G**: FISH performed at 56 hpf with the *pax6b* probe on wild-type embryos without (A) or with the transgenes indicated on the Fig (B-G) and heat-shocked at 38 and 48 hpf. **H:** Quantification using Imaris software of the volume occupied by the induced *pax6b+* cells in the gut for the conditions A to G with Tg2 (Neurod1), arbitrarily set to 100%. All Views are ventral with the anterior part to the left and represent confocal projection images. The pancreas is encircled while the location of the gut, visualised with a DAPI staining

(not shown), is delimited by dashed lines. Scale bar: 100μm.
(TIF)

**S5 Fig. Gene Ontology analysis of _neurod1_$^{ΔUCE}$ DE genes.** Gene Ontology analysis highlights significant enrichment for insulin signaling pathway in _neurod1_$^{ΔUCE}$ DE genes. 13 out of 60 genes of the cascade are differentially expressed (indicated by red dots).
(TIF)

**S1 Table. List of the DE genes for Ascl1a, Neurod1, Neurod1$^{ΔECD}$ and Neurod1$^{ΔUCE}$ GOF.** Expression level of the DE genes given in CPM (counts per million) for the 16 samples as well as the mean and the FDR. **Sheet A**: DE genes for Neurod1 versus ctrl; **Sheet B**: DE genes for ascl1a versus ctrl; **Sheet C**: DE genes common for Ascl1a and Neurod1; **Sheet D**: DE genes for ascl1a versus Neurod1; **Sheet E**: DE genes for Neurod1$^{ΔUCE}$ and Neurod1$^{ΔECD}$ versus Neurod1. The list of the expression level of all genes at 52 hpf is provided in S2 Table.
(XLSX)

**S2 Table. Expression level at 52 hpf of all genes given in CPM for the 16 samples with the means.**
(XLSX)

**S3 Table. List of the DE genes for the Neurod1$^{ΔUCE}$ LOF.** Expression level of the DE genes (FDR ≤ 0,1, average expression level >10 CPM) given in CPM for the 6 ctrl and 4 Neurod1$^{ΔUCE}$ EEC as well as the mean and the FDR. **Sheet A**: List of the 680 DE genes; **Sheet B**: List of the 75 transcription factors DE.
(XLSX)

**S4 Table. List of the primers used for constructing the transgene, the Crispr/cas9 guides and the T3 flagged probes.**
(XLSX)

**S5 Table. Numerical data for all graphs.**
(XLSX)

## Acknowledgments

We thank Bruce Riley for the Tg(hsp70:atoh1a) line [15], Sudipto Roy for the Tg(Hsp70: atoh1b-Myc) line [14] and Stephanie Woo for the tg(sox17:dsred)$^{s903}$ line [64]. We also acknowledge M Hammerschmidt's lab for providing the pia mutants [42] and Damien Polys and Didier Cataldo for the scanning of the histological sections. We thank the following technical platforms: GIGA-Zebrafish (H Pendeville), GIGA-Cell Imaging and Flow Cytometry (S Ormenese and S Raafat), the GIGA-Genotranscriptomic (W Coppieters, L Karim and M. Deckers) and the GIGA-Immunohistology platforms.

## Author Contributions

**Conceptualization:** Anne Sophie Reuter, David Stern, Alice Bernard, Lydie Flasse, Bernard Peers, Marianne L. Voz.

**Data curation:** Arnaud Lavergne.

**Formal analysis:** David Stern, Chiara Goossens, Arnaud Lavergne, Marianne L. Voz.

**Funding acquisition:** Marianne L. Voz.

**Investigation:** Anne Sophie Reuter, David Stern, Alice Bernard, Chiara Goossens, Lydie Flasse, Virginie Von Berg, Marianne L. Voz.

**Methodology:** Anne Sophie Reuter, David Stern, Alice Bernard, Chiara Goossens, Lydie Flasse, Virginie Von Berg.

**Project administration:** Anne Sophie Reuter, Marianne L. Voz.

**Software:** Arnaud Lavergne.

**Supervision:** Isabelle Manfroid, Bernard Peers, Marianne L. Voz.

**Validation:** Anne Sophie Reuter, David Stern, Alice Bernard, Virginie Von Berg, Marianne L. Voz.

**Visualization:** Anne Sophie Reuter, Lydie Flasse, Marianne L. Voz.

**Writing – original draft:** Anne Sophie Reuter, Bernard Peers, Marianne L. Voz.

**Writing – review & editing:** Anne Sophie Reuter, Isabelle Manfroid, Bernard Peers, Marianne L. Voz.

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
