## [Decision Letter · Decision Letter 0]

3 Jun 2021

Dear Dr Voz,

Thank you very much for submitting your Research Article entitled 'Identification of an evolutionarily conserved domain in Neurod1 essential for triggering enteroendocrine cell differentiation' to PLOS Genetics.

The manuscript was fully evaluated at the editorial level and by independent peer reviewers. The reviewers appreciated the attention to an important problem, but raised some substantial concerns about the current manuscript. Based on the reviews, we will not be able to accept this version of the manuscript, but we would be willing to review a much-revised version. We cannot, of course, promise publication at that time.

As you will see, a main concern is the lack of data describing the phenotype of Neurod1-null mutant larva precluding definitive conclusions on the role of this TF and particularly of its UCE domain in the differentiation of the intestinal secretory lineage. Additional time (6 months instead of 60 days) to complete the experiments and resubmit has been allowed (see below).

If you decide to revise the manuscript for further consideration at PLOS Genetics, please aim to resubmit within the next 6 months, unless it will take extra time to address the concerns of the reviewers, in which case we would appreciate an expected resubmission date by email to plosgenetics@plos.org.

[LINK]

We are sorry that we cannot be more positive about your manuscript at this stage. Please do not hesitate to contact us if you have any concerns or questions.

Yours sincerely,

Gérard Gradwohl, Ph.D

Guest Editor

PLOS Genetics

Gregory Barsh

Editor-in-Chief

PLOS Genetics

Reviewer's Responses to Questions

**Comments to the Authors:**

Reviewer #1: The authors investigate whether different members of the achaete-scute family of basic helix loop helix transcription factors are able to restore loss of secretory cells in ascl1a mutants. They find that all members of the family are able to rescue secretory cells. Each family member does not rescue all classes secretory cells. Neurod1 is the is the only family member able to rescue enteroendocrine cells. The other family members rescue goblet cells but not enteroendocrine. Neurod1 has a conserved region that is able to induce enteroendocrine fates and repress goblet cell fate. Transcriptomics of ectopically expressed neurod1 and varients upregulate a group of differentially expressed genes and ascl1a downregulates many of these genes. There is a common group of differentially expressed genes shared by neruod1 and ascl1a.

The authors do a good job on analysis of the activity of ectopically expressed achaete-scute family members. The activity of each of these family members is nicely characterized in terms of secretory cell formation and differential gene expression.

Regarding ectopic expression of achaete-scute family members, the authors demonstrate misexpression of secretory cell markers and genes within the intestine but do not perform histology to show whether ectopic expression alters the overall structure of the embryo and more specifically the structure of the intestine. Authors should show images of whole embryo and histology of intestine to demonstrate whether structure is normal other than changes in secretory cell content.

Authors should more specifically describe controls for heat shock experiments. It was hard to find description of controls in either in results or material and methods.

Reviewer #2: In the present manuscript, Reuter et al address the specificity of action of a family of closely related basic Helix-Loop-Helix (bHLH) containing transcription factors in the establishment of secretory lineages in the developing zebrafish larval gut. To achieve this, the authors use a rescue approach based on a series of global misexpression transgenic constructs in a mutant background, ascl1a/pia loss-of-function, that abrogates all secretory linages. In this manner, they find that while Ascl1a/b, Atoh1a/b and Neurod1 factors all drive cells in the gut to differentiate, the specific identity of the neurons that are generated varies depending on the factor used – the former two families rescue goblet cells and Neurod1 rescues enteroendocrine (EE) cells. Using a subsequent series of transgenes driving chimeric bHLH factors, the authors identify a region of Neurod1 responsible for its EE cell specificity. Deletion of this region using Crispr/Cas9 genome editing partially abrogates this specific function. Finally, a series of transcriptome analyses highlights potential target lists that are downstream of these bHLH factors.

Overall, the manuscript presents a considerable sum of data, including RNAseq data. Unfortunately, the relatively simple message the authors try to convey via the title or abstract is not completely supported.

My major and minor comments are as follows:

Major comments:

The authors construct a gene regulatory pathway suggesting that Ascl1a lies upstream of Neurod1, which in turn inhibits Gfi1aa leading to inhibition of goblet cell fate, and that this pathway hinges on the presents of the Ultra-conserved element (UCE) in Neurod1. Unfortunately, the evidence for this is largely anecdotal and is not directly tested.

1) To start, the inability of Ascl1a misexpression to completely rescue the ascl1a mutant phenotype is perturbing. The authors present various arguments as to why this might be the case. Nonetheless, it severely biases the possible interpretations of the rescue results for Neurod1 as it begs the question of whether what neurod1 can do in misexpression reflects its true function – Neurod1 can do certain things whether it does them endogenously is unclear.

2) The phenotype of neurod1-null mutant larvae needs to be presented. While the UCE mutant established by the authors is intriguing, without the neurod1-null, it is not easy to interpret the UCE-/- phenotype. If the null and UCE mutations show the same phenotype, for instance, it becomes difficult to conclude a specific function for the UCE.

3) The authors speculate that the inability of Ascl1a to rescue the EE lineage is due to its inability to restore neurod1 expression. A key element that could have helped the authors arguments would have been to know if misexpression of Neurod1 can rescue in the neurod1-null background.

4) The authors speculate about Neurod1/Gfi1aa interactions, but to my knowledge no role for Gfi1aa has been established in the zebrafish larval gut.

5) The RNAseq data may prove to be a useful resource for future studies. As these datasets are based on misexpression, however, it is difficult to assess whether they reflect the endogenous activities of the various bHLH factors. As these datasets neither support nor contradict the model presented in the abstract, they could be removed without altering the interpretation the authors present concerning the Ascl1a>Neurod1>Gfi1aa pathway.

Minor comments:

1) The distribution of the various markers along the A/P axis of the gut appears to vary as a function of the bHLH used in the rescue experiments. This is more clearly the case for sox4b expression (panels C,D,E). Is this significant?

2) Figure 2 would benefit from having wildtype controls presented as in Figure 1.

3) It is not clear why the activity of Tg4 and 8 were not determined concerning pax6b induction? What does “V” indicate in Table 1?

4) The distinction between full and partial rescue presented in Figure 4F is not specified.

5) Advancing the time window for the RNAseq analyses such that it precedes the endogenous induction of pax6b does not really “circumvent” the problem that ascl1a mutants cannot be distinguished from wildtype or heterozygote siblings. It does, however, minimize some of the risks inherent in these experiments.

6) The images in Figure 6 are not of sufficient quality and should be replaced.

7) Figure 7 would benefit from high magnification. As cell counts are presented, it would be nice to see that cells are sufficiently distinct to be counted and the low magnification images presented do not permit this.

8) There are problems throughout the manuscript with gene/protein/transgene nomenclature that need to be corrected.

9) The authors should stick to either “EE cells” or “EEC”.

10) Not all panels of Figures are referred to in the text – Figure 4A, for instance. Some panels are referenced but do not exist – Figure 4H, for instance. “Fig” versus “Figure” versus “figure” needs to be homogenized.

**Have all data underlying the figures and results presented in the manuscript been provided?**

Reviewer #1: Yes

Reviewer #2: **No: **The RNAseq data is not yet available apparently.

PLOS authors have the option to publish the peer review history of their article (what does this mean?). If published, this will include your full peer review and any attached files.

Reviewer #1: No

Reviewer #2: No

---

## [Decision Letter · Decision Letter 1]

19 Jan 2022

Dear Dr Voz,

Thank you very much for submitting your Research Article entitled 'Identification of an evolutionarily conserved domain in Neurod1 favouring enteroendocrine versus goblet cell fate' to PLOS Genetics.

The manuscript was fully evaluated at the editorial level and by independent peer reviewers. 

Generation and analysis of the NeuroD mutant, requested by Reviewer 2, significantly reinforces the conclusion on the role of this TFs in lineage determination in the zebrafish intestine.  Reviewer 1 raises some concerns on the difficulty to interpret the data regarding secretory lineage allocation given that the overall intestinal epithelium structure looks altered (at least in the UCE mutant) based on the novel histological analysis  he requested and that you incorporated in figure 8 . We therefore ask you to provide additional  explanations or data that could  dissipate this concern.

[LINK]

Yours sincerely,

Gérard Gradwohl, Ph.D

Guest Editor

PLOS Genetics

Gregory Barsh

Editor-in-Chief

PLOS Genetics

Reviewer's Responses to Questions

**Comments to the Authors:**

Reviewer #1: Authors add histology for the neuroD1 mutant but fail to add histology from any of the experiments using the transgenic fish. Determination of whether the intestine is organized in a similar way to WT is important for interpretation of experiments.

Histology added to Figure 8 for the neuroD1 mutant makes it hard to determine how the epithelium is affected. Is the epithelium still a single layer of cells in the region with additional goblet cells? It looks like the lumen may be occluded or the epithelium is stratified.

Reviewer #2: The authors have addressed the major point raised concerning the null phenotype of a neuroD1 mutant and the results are clear. As such, I have no further issues with the study.

**Have all data underlying the figures and results presented in the manuscript been provided?**

Reviewer #1: Yes

Reviewer #2: Yes

PLOS authors have the option to publish the peer review history of their article (what does this mean?). If published, this will include your full peer review and any attached files.

Reviewer #1: No

Reviewer #2: No

---

## [Editor Report · Decision Letter 2]

17 Feb 2022

Dear Dr VOZ,

We are pleased to inform you that your manuscript entitled "Identification of an evolutionarily conserved domain in Neurod1 favouring enteroendocrine versus goblet cell fate" has been editorially accepted for publication in PLOS Genetics. Congratulations!

Yours sincerely,

Gérard Gradwohl, Ph.D

Guest Editor

PLOS Genetics

Gregory Barsh

Editor-in-Chief

PLOS Genetics

Comments from the reviewers (if applicable):

**Data Deposition**

http://datadryad.org/submit?journalID=pgenetics&manu=PGENETICS-D-21-00533R2

**Press Queries**

---

## [Editor Report · Acceptance letter]

10 Mar 2022

PGENETICS-D-21-00533R2 

Identification of an evolutionarily conserved domain in Neurod1 favouring enteroendocrine versus goblet cell fate 

Dear Dr Voz, 

We are pleased to inform you that your manuscript entitled "Identification of an evolutionarily conserved domain in Neurod1 favouring enteroendocrine versus goblet cell fate" has been formally accepted for publication in PLOS Genetics! Your manuscript is now with our production department and you will be notified of the publication date in due course.

With kind regards,

Livia Horvath

PLOS Genetics

On behalf of:
